# Gut microbiome dysbiosis across early Parkinson's disease, REM sleep behavior disorder and their first-degree relatives

Bei Huang [1,2], Steven W. H. Chau [1,2], Yaping Liu[1,2,3], Joey W. Y. Chan [1,2], Jing Wang [1,2,3], Suk Ling Ma [2], Jihui Zhang [1,2,3], Paul K. S. Chan [4,5], Yun Kit Yeoh [4], Zigui Chen [4,5], Li Zhou [1,2], Sunny Hei Wong[6,7,8], Vincent C. T. Mok [7,9,10], Ka Fai To [5,6,11], Hei Ming Lai [2,7,9], Simon Ng[5,6,12], Claudia Trenkwalder[13,14], Francis K. L. Chan[5,6,7,9] & Yun Kwok Wing [1,2,5] ✉

The microbiota-gut-brain axis has been suggested to play an important role in Parkinson's disease (PD). Here we performed a cross-sectional study to profile gut microbiota across early PD, REM sleep behavior disorder (RBD), first-degree relatives of RBD (RBD-FDR), and healthy controls, which could reflect the gut-brain staging model of PD. We show gut microbiota compositions are significantly altered in early PD and RBD compared with control and RBD-FDR. Depletion of butyrate-producing bacteria and enrichment of pro-inflammatory *Collinsella* have already emerged in RBD and RBD-FDR after controlling potential confounders including antidepressants, osmotic laxatives, and bowel movement frequency. Random forest modelling identifies 12 microbial markers that are effective to distinguish RBD from control. These findings suggest that PD-like gut dysbiosis occurs at the prodromal stages of PD when RBD develops and starts to emerge in the younger RBD-FDR subjects. The study will have etiological and diagnostic implications.

Alpha-synucleinopathies, such as Parkinson's disease (PD), are characterized by the abnormal aggregation of alpha-synuclein (α-syn) protein in the central nervous system (CNS)[1,2]. However, increasing evidences suggested that α-syn pathology has already occurred in the enteric nervous system (ENS) prior to the involvement of CNS[3,4], which strongly supported the gut-to-brain propagation of α-synucleinopathy as proposed in refs. 2,5. In parallel, gut microbiota disturbance (gut dysbiosis), an emerging biomarker and intervention target for various

[1]Li Chiu Kong Family Sleep Assessment Unit, Department of Psychiatry, Faculty of Medicine, The Chinese University of Hong Kong, Hong Kong SAR, China. [2]Department of Psychiatry, Faculty of Medicine, The Chinese University of Hong Kong, Hong Kong SAR, China. [3]Center for Sleep and Circadian Medicine, The Affiliated Brain Hospital of Guangzhou Medical University, Guangzhou, Guangdong, China. [4]Department of Microbiology, Faculty of Medicine, The Chinese University of Hong Kong, Hong Kong SAR, China. [5]Centre for Gut Microbiota Research, Faculty of Medicine, The Chinese University of Hong Kong, Hong Kong SAR, China. [6]Institute of Digestive Disease, State Key Laboratory of Digestive Disease, Faculty of Medicine, The Chinese University of Hong Kong, Hong Kong SAR, China. [7]Department of Medicine and Therapeutics, Faculty of Medicine, The Chinese University of Hong Kong, Hong Kong SAR, China. [8]Centre for Microbiome Medicine, Lee Kong Chian School of Medicine, Nanyang Technological University, Singapore, Singapore. [9]Li Ka Shing Institute of Health Sciences, Faculty of Medicine, The Chinese University of Hong Kong, Shatin, Hong Kong, China. [10]Margaret K.L. Cheung Research Centre for Management of Parkinsonism, Department of Medicine and Therapeutics, Faculty of Medicine, The Chinese University of Hong Kong, Hong Kong SAR, China. [11]Department of Anatomical and Cellular Pathology, Faculty of Medicine, The Chinese University of Hong Kong, Hong Kong SAR, China. [12]Department of Surgery, Faculty of Medicine, The Chinese University of Hong Kong, Hong Kong SAR, China. [13]Clinic for Neurosurgery, University Medical Center, Georg August University Göttingen, Göttingen, Germany. [14]Center of Parkinsonism and Movement Disorders, Paracelsus-Elena Hospital, Kassel, Germany. ✉e-mail: ykwing@cuhk.edu.hk

complex diseases, has been consistently reported in PD patients[6]. It was hypothesized that PD-associated gut dysbiosis, especially the depletion of short-chain fatty acids (SCFA)-producing bacteria[7–9] and enrichment of putative pathobionts[10], was related to intestinal hyperpermeability[11], immune activation[12], and pathological α-syn aggregation[11,13]. Nonetheless, given that enteric α-syn pathology and ENS dysfunctions especially constipation could occur decades before the onset of PD[3,14], it is critical to understand gut microbiota and host–microbiome interactions at much earlier prodromal stages of PD before evident motor symptoms develop.

REM sleep behavior disorder (RBD) is perceived as the most specific prodromal marker of PD, characterized by dream-enactment behaviors and REM sleep without atonia[15]. Patients with video-polysomnography (v-PSG)-confirmed RBD reported an increased prevalence of constipation[16], and increased phosphorylated α-syn immunostaining in ENS[4]. Likewise, PD patients with premotor RBD features appeared to exhibit prominent degeneration of the peripheral nervous system (e.g., increased constipation and enteric α-syn histopathology) when comparing to those without, suggesting a distinct subtype of Parkinson's disease that reflects the gut-brain hypothesis of α-synucleinopathy[17]. On the other hand, isolated RBD symptoms, but not yet meeting the v-PSG diagnostic criteria for RBD, might reflect a prodromal stage of RBD and the early presentation of α-synucleinopathy[18,19]. A recent case–control–family study reported that the first-degree relatives of RBD (RBD-FDR) had increased constipation and a spectrum of RBD features: from isolated RBD symptoms (indicative of prodromal RBD) to v-PSG-diagnosed RBD. Therefore, RBD-FDR might harbor a group of susceptible individuals at a much earlier stage of α-synucleinopathy than RBD patients[20].

Prior studies have reported gut microbiota dysbiosis in v-PSG diagnosed RBD ($n = 21$ and 26, respectively)[21,22] and possible RBD as assessed by screen questionnaire ($n = 84$)[23]. However, although these prior studies suggested a similar trend of changes in microbial composition in RBD and PD, they might be underpowered to comprehensively detect host–microbiome interactions. In addition, a prodromal stage of RBD has been increasingly recognized[24], underscoring the importance of studying gut microbiota at an even earlier prodromal stage. Here, we performed a large cross-sectional study across prodromal and early stages of disease (i.e., simulate the Braak staging model with a quasi-longitudinal design)[2], to disentangle the associations of gut microbiota with the progression of α-synucleinopathy.

## Results

### Sociodemographic and clinical characteristics

This study includes stool samples from 452 subjects from the cohorts of v-PSG-diagnosed RBD and RBD family in Hong Kong. After excluding 11 samples with low read count, a total of 441 samples remained for further analyses (Fig. 1). All patients with early PD had clinically confirmed PD with motor symptoms onset less than 5 years. Control ($n = 108$, $67.3 \pm 7.0$ years, 63.9% males) were age- and sex-matched with RBD ($n = 170$, $68.6 \pm 7.6$ years, 73.5% males) and early PD groups ($n = 36$, $67.8 \pm 5.6$ years, 86.1% males). All three groups were older with more males when comparing to RBD-FDR ($n = 127$, $q$ values <0.05). The severity of RBD features, as captured by RBDQ-HK questionnaire, was significantly increased from control, RBD-FDR to RBD and early PD (total score of RBDQ-HK, $6.3 \pm 7.0$ vs $9.2 \pm 8.4$ vs $39.2 \pm 17.7$ vs $32.8 \pm 16.1$, $P$ value <0.001). Of 127 RBD-FDR, 11 (8.7%) were diagnosed with probable RBD based on a structured clinical interview[25]. Total likelihood ratio (LR) for prodromal PD is a research criterion used to identify subjects at risk of having prodromal PD. We found that RBD patients had greater LR of prodromal PD (excluding RBD item) than control (log-transformed LR, $1.4 \pm 0.98$ vs $0.58 \pm 0.72$, $q$ value <0.001) and RBD-FDR ($1.4 \pm 0.98$ vs $0.46 \pm 0.55$, $q$ value <0.001), while control and RBD-FDR had comparable levels of total LR (Supplementary Dataset 1).

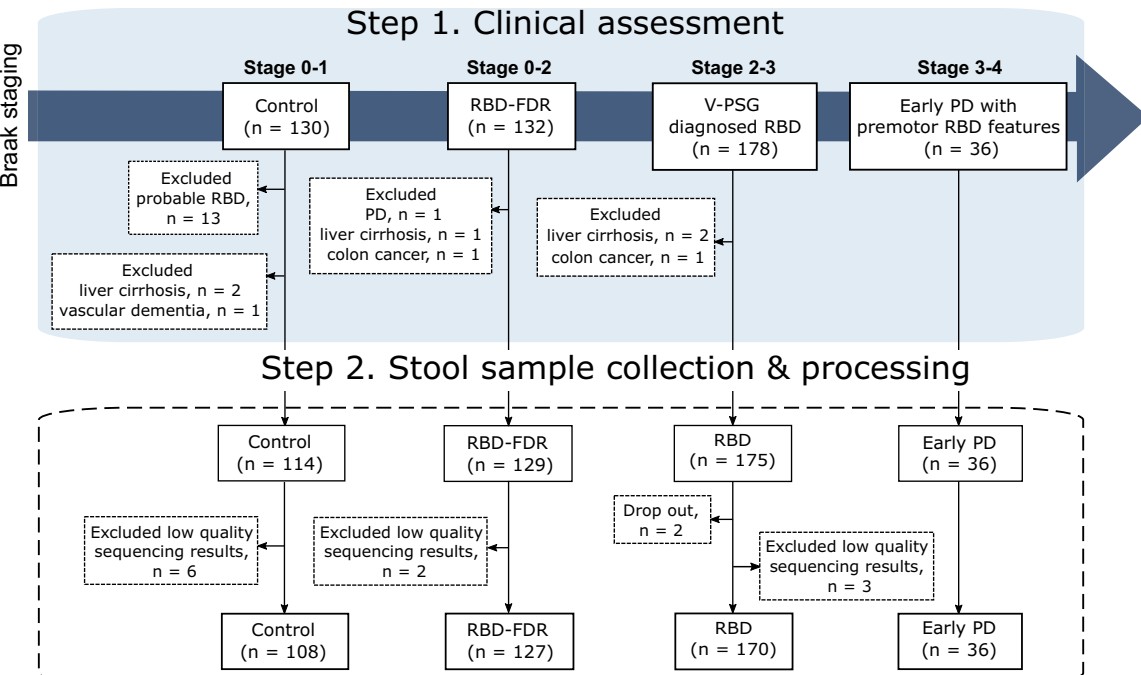

**Fig. 1 | Flowchart of subject selection and recruitment.** We recruited subjects according to the proposed staging model of α-synucleinopathy, which aptly represented the pathological staging of Parkinson's disease (i.e., Braak staging). Four different clinical stages were controls (Braak stage 0–1), RBD-FDR (stage 0–2), patients with RBD (stage 2–3) and early PD (stage 3–4). Early PD refers to patients who had clinically confirmed PD with motor symptoms onset less than 5 years. Control subjects with probable RBD as diagnosed by using structured clinical interview were excluded. Besides, subjects with neurodegenerative diseases (except early PD group) and severe gastrointestinal diseases were excluded from this study. In the end, a total of 452 subjects successfully collected stool samples, while 11 of them were removed for subsequent analysis due to the low quality of sequencing data (i.e., total read count <1000). RBD REM sleep behavior disorder, RBD-FDR first-degree relatives of patients with RBD, PD Parkinson's disease.

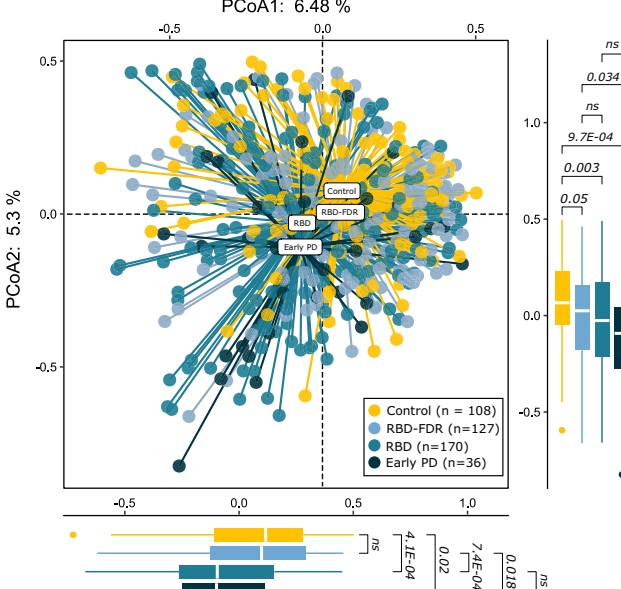

**a** Shifted microbial composition at prodromal and early α-synucleinopathy

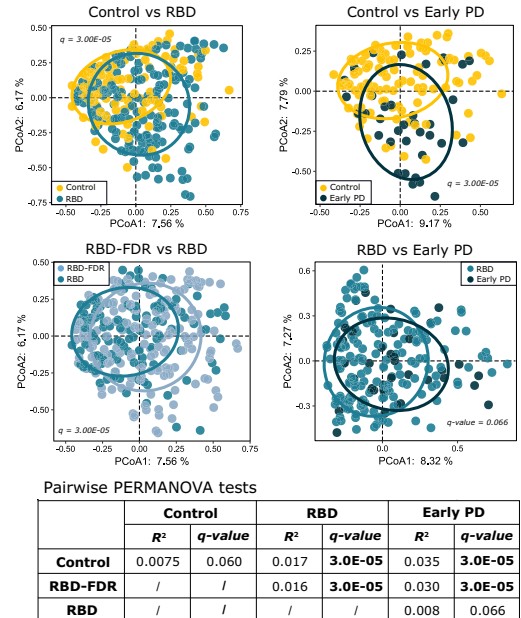

**b** Compositional differences in gut microbiota between the groups

**Fig. 2 | Shifted microbial composition across the prodromal and early stages of α-synucleinopathy. a** Principal coordinates analysis (PCoA) of microbial communities across control ($n = 108$), RBD-FDR ($n = 127$), RBD ($n = 170$), and early PD ($n = 36$) based on Bray–Curtis distance matrix at the genus level. The label of each group indicates group centroid. Boxplots along the axes of PCoA showed the distributions of PCoA1 and PCoA2 between groups. The white line in the box represented median values, while the lower and upper boundaries represented the first and third quartiles, respectively; whiskers extend up to values within 1.5 times of the interquartile range; outliers are plotted as individual points beyond the whiskers. Statistical differences were analyzed using ANOVA (two-sided test) with post hoc test. *P* values for multiple testing were adjusted by applying

Benjamini–Hochberg method. **b** Principal coordinates analysis of microbial communities between groups with 70% confidence ellipse. The significance of intergroup differences in overall microbial composition was calculated by PERMANOVA with adjustment of age and sex (permutation = 99,999, two-sided test, based on Bray–Curtis distance matrix at the genus level). $R^2$ indicated the inter-individual variation explained by grouping factors in PERMANOVA test, and *P* values for pairwise comparisons were adjusted by applying Benjamini–Hochberg method. RBD REM sleep behavior disorder, RBD-FDR first-degree relatives of patients with RBD, PD Parkinson's disease, PERMANOVA permutational multivariate analysis of variance, ns not significant. Source data are provided as a Source Data file.

Gastrointestinal symptoms were assessed by the Rome-IV diagnostic questionnaire for adults and Scales for Outcomes in Parkinson's Disease–Autonomic[26,27]. The prevalence of functional constipation showed an increasing trend from control, RBD-FDR, RBD to early PD patients (8.3 vs 9.4 vs 45.3 vs 69.4%, *P* value <0.001). Straining with defecation, a core feature of functional constipation, progressively increased across four groups even after adjusting age and sex (8.8 vs 15.8 vs 45.4 vs 68.6%, *P* value <0.001). Besides, we used bowel movement frequency score [BMF, ranges from 1 ("bowel movement >1/day") to 6 ("≤1/week")] and stool consistency (inverse scoring of the Bristol Stool Form Scale [BSFS], with higher scores indicating harder stools) as the proxies for colon transit time, both of which demonstrated increasing trends across the four groups (*P* values <0.001). Other gastrointestinal disorders, such as irritable bowel syndrome and functional diarrhea, did not differ among the groups.

In terms of clinical characteristics, RBD patients reported more lifetime major depressive disorder and anxiety disorders than control and RBD-FDR (all *q* values <0.05)[28]. Other potential confounding human diseases, such as diabetes and dyslipidemia, were similarly distributed among four groups. Medication usage referred to any drugs taken during the period of stool collection. It was found that more than half of the RBD and early PD patients took benzodiazepines, while 30.6% and 13.9% of early PD patients and 5.3% and 25.3% of RBD patients were taking osmotic laxatives and antidepressants, respectively. As for PD-specific drugs, 47.2% early PD patients received carbidopa/levodopa, followed by monoamine oxidase B inhibitors (41.7%), dopamine agonist (8.3%), benzhexol hydrochloride (5.6%), and

catechol-O-methyltransferase inhibitors (2.8%). Lifestyle features, including pre-/probiotics consumption and subjective physical activity, showed no significant differences between groups (Supplementary Dataset 1).

### Shifted gut microbiota composition with the progression of α-synucleinopathy
A total of 84 families and 249 genera were recognized from 441 fecal samples. We observed that alpha diversity (Chao 1, Gini Simpson, and Shannon indexes) at genus levels were comparable between groups (Supplementary Dataset 2). Inter-individual dissimilarity of microbiota composition (i.e., beta-diversity) was assessed between each pair of groups using permutational multivariate analysis of variance (PERMANOVA, 99,999 permutations) with adjustment of age and sex (Supplementary Dataset 3). The early PD group presented a distinct clustering pattern of microbiota relative to the control ($R^2 = 0.035$, *q* value <0.001). Similarly, the microbiota composition of RBD was akin to that of early PD ($R^2 = 0.008$, *q* value = 0.066), but significantly differed from control and RBD-FDR (all *q* values <0.001). The microbial compositional analysis revealed no significant differences between control and RBD-FDR (Fig. 2b). Homogeneity of dispersion test indicated that RBD and early PD had higher levels of variation (all *q* < 0.05, Supplementary Fig. 2a), which may potentially affect PERMANOVA results, especially for the early PD group (smaller number of subjects)[29]. Nevertheless, the gut microbiota composition of early PD versus RBD and control would still be supported by the inter-group comparisons along principal components as well as the distinct

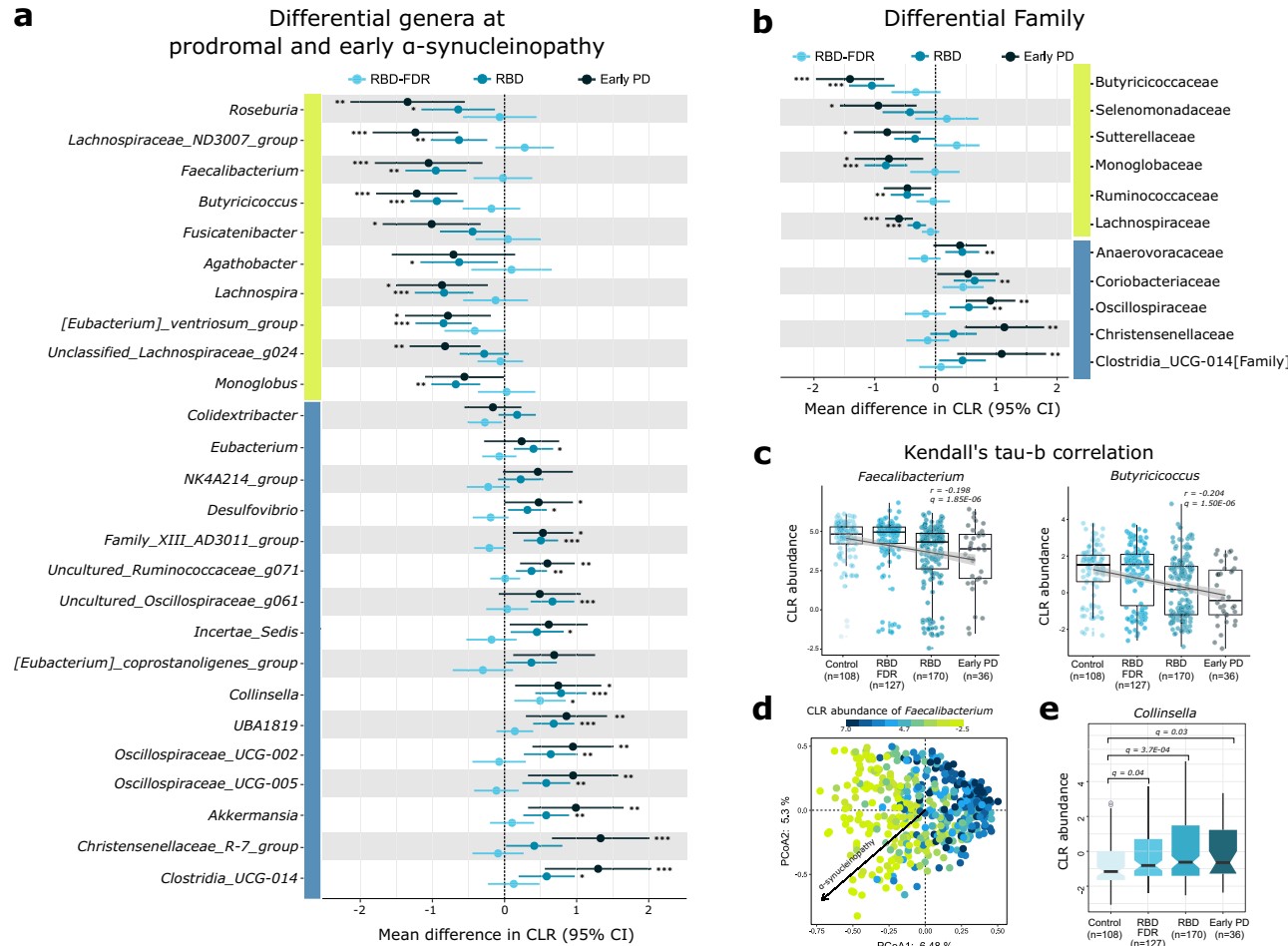

**Fig. 3 | Differential gut microbes at the prodromal and early stages of α-synucleinopathy. a, b** Error bar plot demonstrated mean difference of CLR-transformed abundance of taxa at prodromal and early α-synucleinopathy as compared with control, and 95% confidence interval of the mean difference. The blue and yellow bars along the vertical axis of the plot indicated taxa increased and decreased with disease progression (Kendall's τb > 0 and <0, respectively). The associations of differentially abundant taxa with the progression of α-synucleinopathy were analyzed using MaAsLin 2. The model was computed with group as fixed effect, family clustering as random effects. "****", "**", and "*" represented Benjamini–Hochberg method adjusted *P* values (*q* value) less than 0.001, 0.01, and 0.05, respectively. At family level, only taxa that significantly changed at prodromal and early α-synucleinopathy was presented, see also Supplementary Fig. 4 and Dataset 8. **c** Kendall's tau-b correlation analysis showed that *Butyricicoccus* and *Faecalibacterium* had strongest correlations with the progression of α-synucleinopathy. Individual data points were shown with the boxplot covers the clustering patterns in principal coordinates (PCoA) analysis, as shown

interquartile interval of the data; median of the data was shown as a thick line in the middle of the box; whiskers extend up to values within 1.5 times of interquartile range. **d** Genus *Faecalibacterium* best explained the variation of first principal component from the principal coordinates analysis of whole genera. Black arrow indicated the influence of disease progression (i.e., from control, RBD-FDR, RBD to early PD). **e** Genus *Collinsella* was progressively increased from control (*n* = 108), RBD-FDR (*n* = 127), RBD (*n* = 170) to early PD (*n* = 36) in MaAsLin 2 (two-sided test, see also Supplementary Dataset 8). The notched boxplot panel showed CLR-transformed abundance. The thick line in box represented median values, while the lower and upper boundaries represented the first and third quartiles, respectively; whiskers extend up to values within 1.5 times of the interquartile range; outliers are plotted as individual points beyond the whiskers. MaAsLin 2 Microbiome Multivariable Associations with Linear Model, CLR centered log ratio, RBD REM sleep behavior disorder, RBD-FDR first-degree relatives of patients with RBD, PD Parkinson's disease. Source data are provided as a Source Data file.

in Fig. 2 and supplementary Fig. 2c, d respectively.

## Gut microbes progressively changed across prodromal and early stages of α-synucleinopathy

After filtering out low prevalence (<10%) and abundance (<0.05%) taxa, a total of 36 families and 88 genera remained for differential abundance analysis. Read count table of genera was reframed to compositional data with centered log-ratio (CLR) transformation. We found that 35.2% (*n* = 31) of genera were significantly associated with four stages of α-synucleinopathy (Kendall's tau-b correlation, *q* value <0.05, Supplementary Dataset 5). Among these genera, *Butyricicoccus* and *Faecalibacterium* showed the strongest correlation with disease progression (Kendall's τb = −0.204 and −0.198, *q* value <0.001, Fig. 3c), while *Faecalibacterium* was also a major contributor to the variation of

first principal component as shown in Fig. 3d (Spearman *r* = 0.72, *P* value = 6.4 × 10⁻⁶⁹, Supplementary Dataset 6).

Differential taxa were assessed using Kruskal–Wallis test at which we identified that 16 families and 26 genera significantly differed among four groups (*q* value <0.05, Supplementary Datasets 5 and 7). Microbiome multivariable associations with linear model (MaAsLin 2) was applied to further estimate the associations of differential taxa across control, RBD-FDR, RBD and early PD. The model included four stages (reference = control) as fixed effect and family id as a random effect. Nineteen (45.2%) out of 42 differential taxa, including butyrate-producing bacteria (e.g., *Roseburia*, *Lachnospiraceae_ND3007_group*, *Lachnospira*, *[Eubacterium]_ventriosum_group*, *Butyricicoccus*, *Faecalibacterium*, and family Lachnospiraceae), hydrogen sulfide-producing *Desulfovibrio*, mucin-degrading *Akkermansia*, *Collinsella*, *Oscillospiraceae_UCG-002* and −*005*, were significantly and similarly

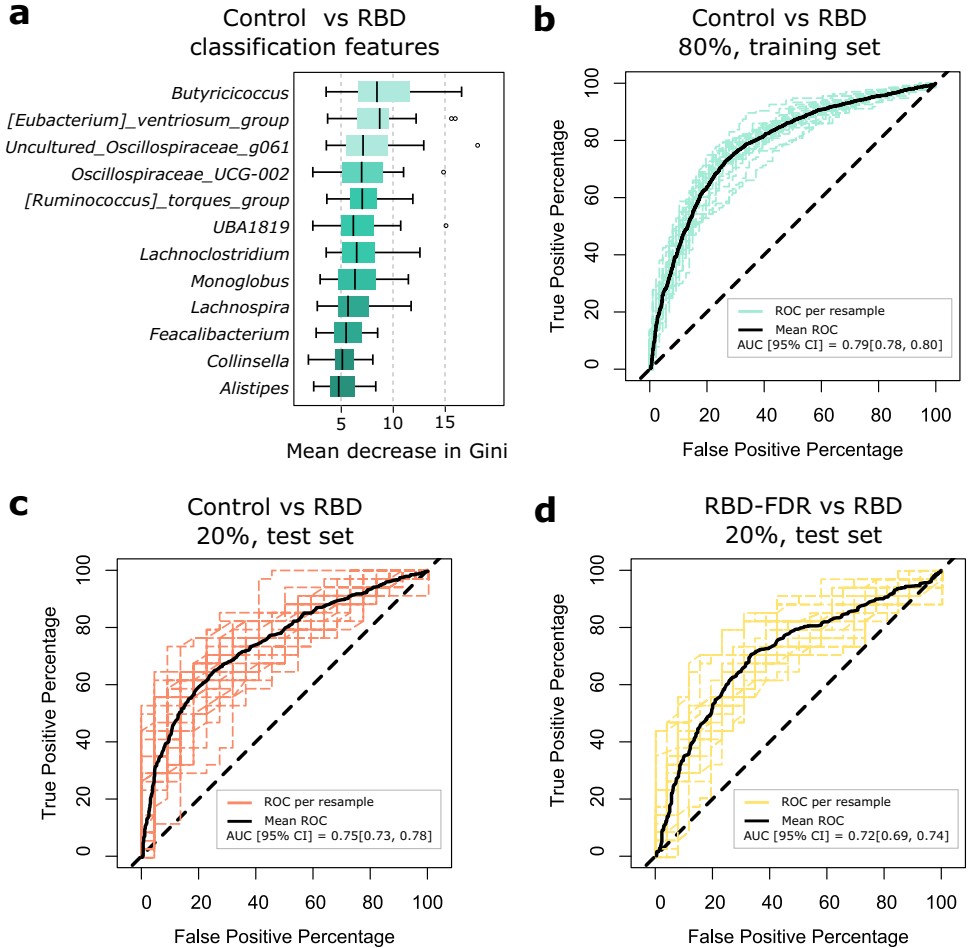

**Fig. 4 | Random Forest model predicting RBD status. a** Feature selection was based on the recursive feature elimination (RFE) algorithm, and microbial markers appeared in at least 60% of all 25 final trained models (classifying RBD [$n = 170$] and control [$n = 108$]) were considered as classification features. Thick line in box represents the median values, while the lower and upper boundaries represent the first and third quartiles, respectively; whiskers extend up to values within 1.5 times of interquartile range; outliers are plotted as individual points beyond the whiskers.

**b–d** The receiver operating characteristic curves and mean AUC (with 95% confidence intervals) of random forest classification models. Dashed curves represented the results from 25 repeats of the whole process of random forest-RFE (i.e., 25 resamples), with bold curves showing the mean performance. RBD REM sleep behavior disorder, ROC operating characteristic curve, AUC area under the ROC curve. Source data are provided as a Source Data file.

altered in RBD and early PD when comparing to the control (unadjusted MaAsLin 2, all $q$ values <0.05, Fig. 3a, Supplementary Dataset 8). These associations remained significant at $q$ value <0.1 when further included age and sex as fixed effects in MaAsLin 2 models (adjusted MaAsLin 2, Supplementary Dataset 9).

Interestingly, the enrichment of pro-inflammatory *Collinsella* has already emerged in RBD-FDR, an earlier prodromal stage of α-synucleinopathy, in both adjusted ($\beta = 0.58$, $q$ value = 0.035) and unadjusted ($\beta = 0.49$, $q$ value = 0.038) models (Fig. 3a, e and Supplementary Datasets 8 and 9). Also, we observed a marginal decrease of butyrate-producing *[Eubacterium]_ventriosum_group* in RBD-FDR as compared to the control group ($\beta = -0.54$, $q$ value = 0.069). Further analysis showed that RBD-FDR with probable RBD ($n = 11$) appeared to have a more pronounced decrease in *[Eubacterium]_ventriosum_group* than those without probable RBD features (CLR abundance, $-0.36 \pm 1.8$ vs $0.35 \pm 1.7$, $q$ value = 0.028, Supplementary Fig. 3).

**Gut microbiota as a potential diagnostic biomarker for RBD**
Based on the microbial changes in RBD patients, we further studied the prediction of RBD status by using microbial markers. The machine learning model was built with a random forest algorithm in a training set (80% of whole database), and the performance of the trained model

was tested in the remaining data (i.e., test set). Matrix of features consisted of CLR abundance of 88 filtered genera, and feature selection was based on the recursive feature elimination (RFE) algorithm via 25 repeats of tenfold cross-validation (Supplementary Fig. 5). In the training set, microbial markers could differentiate RBD from control with a mean area under the receiver operating characteristic curve (AUC) of 0.79 ([95% CI] = [0.78, 0.80], Fig. 4b). For the independent test set, the mean accuracy of predictive model was 0.68 ([95% CI] = [0.66, 0.70]), with an AUC of 0.75 ([95% CI] = [0.73, 0.78], Fig. 4c). Feature selection showed that 12 out of 88 genera appeared in at least 60% (15/25) of final feature set of trained models. Among them, genus *Butyricicoccus*, *UBA1819*, *Lachnoclostridium*, *Oscillospiraceae_UCG-002*, *Uncultured_Oscillospiraceae_g061*, *[Ruminococcus]_torques_group*, and *[Eubacterium]_ventriosum_group* were present in all 25 final models for classification of control and RBD (Supplementary Dataset 16).

Gut microbiota-based classifier also showed good performance in distinguishing RBD from RBD-FDR, yielding a mean accuracy of 0.67 ([95% CI] = [0.66, 0.69]) and AUC of 0.72 ([95% CI] = [0.69, 0.74], Fig. 4d) in the test set. The results of feature selection in this model were shown in the Supplementary Fig. 6, with genera *Family_XIII_AD3011_group*, *Uncultured_Oscillospiraceae_g061*, *Faecalibacterium*, *Butyricicoccus*, and *Oscillospiraceae_UCG-002* as the best five classification features.

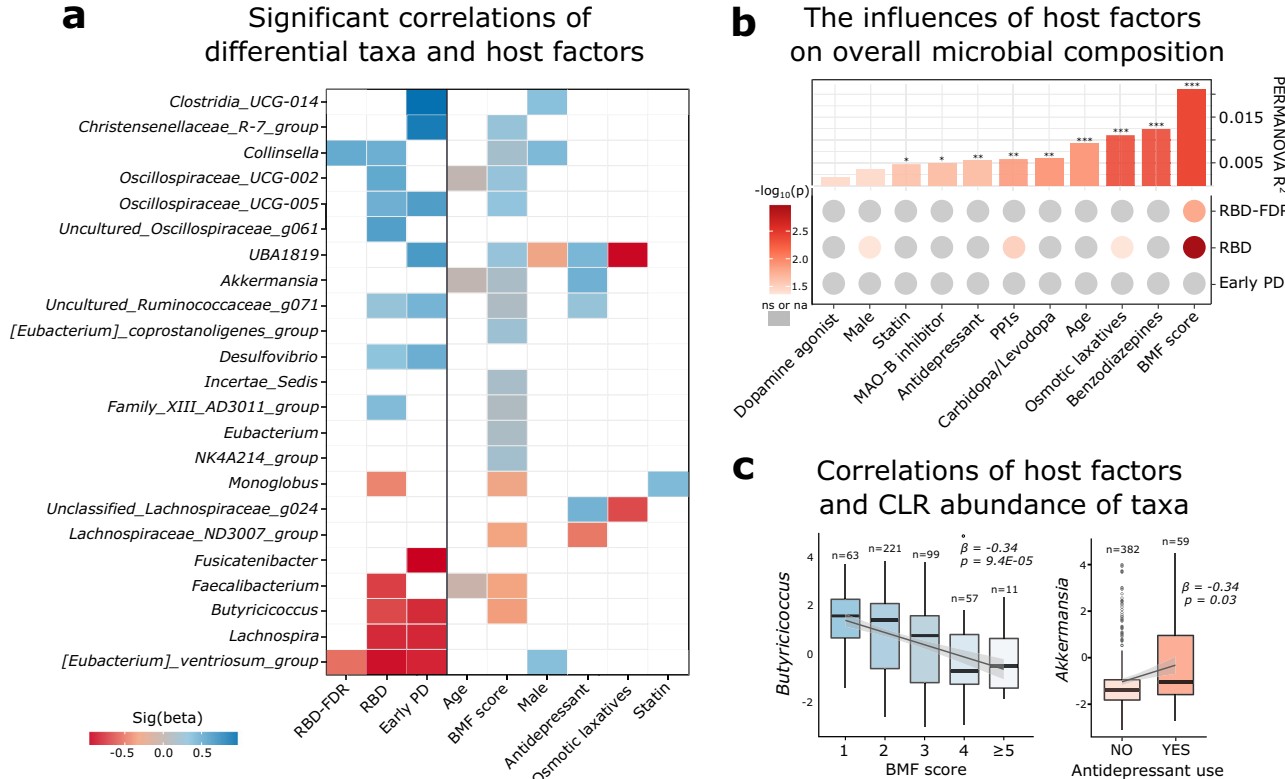

**Fig. 5 | Host–microbiome interactions at prodromal and early α-synucleinopathy.**
**a** The associations between taxa abundance and host factors were assessed by using MaAsLin 2. Only significant results were shown in the heatmap (Benjamini–Hochberg adjusted $P$ values <0.25). For detailed descriptions and results of MaAsLin 2, see Supplementary Dataset 10. **b** Interactions of overall microbial composition and host factors. Boxplot displayed the inter-individual variation explained by each host factor in whole samples by using PERMANOVA test (i.e., PERMANOVA $R^2$, permutations = 99,999). "***", "**", and "*" represented Benjamini–Hochberg adjusted $P$ values <0.001, <0.01, and <0.05, respectively. Heatmap showed the stage-specific (i.e., each stage versus control) impact of host factors. All significant associations were highlighted with red color ($q$ value <0.05 from the PERMANOVA model). See also Supplementary Dataset 4. **c** MaAsLin 2 test revealed significant correlations between BMF score and short-chain fatty acids-

producing bacteria *Butyricicoccus*, as well as the co-occurrence of antidepressant use and RBD/early PD-enriched genera *Akkermansia*. Beta and $P$ value (without Benjamini–Hochberg adjustment) were derived from MaAsLin 2 (two-sided test, see also Supplementary Dataset 10). The thick line in box represents the median values, while the lower and upper boundaries represent the first and third quartiles, respectively; whiskers extend up to values within 1.5 times of the interquartile range; outliers are plotted as individual points beyond the whiskers. RBD REM sleep behavior disorder, RBD-FDR first-degree relatives of patients with RBD, PD Parkinson's disease, PERMANOVA permutational multivariate analysis of variance, BMF bowel movement frequency, PPIs proton pump inhibitors, CLR centered log ratio, MaAsLin 2 Microbiome multivariable associations with linear model, ns not significant, na not applicable. Source data are provided as a Source Data file.

## Host factors exert effects on the microbiota at prodromal and early α-synucleinopathy

Furthermore, we examined the impact of host factors on microbial changes at different stages of α-synucleinopathy. Sociodemographic (age and sex), BMF score, and medications (including antidepressants and benzodiazepines) that exhibited confounding effects on microbiota were evaluated. Psychiatric disorders were not included as covariates since they were highly correlated with antidepressant usage (Chi-squared test, $P$ value = $5.3 \times 10^{-22}$). Multivariate statistical models (PERMANOVA, 99,999 permutations) were built including each pair of groups and all covariates, while PD-specific drugs (Carbidopa/Levodopa, MAO-B inhibitors, and dopamine agonist) were only assessed in the comparisons involving early PD group. In the current model, we still observed a significant compositional shift of microbiota in RBD and early PD as compared with control (all $q$ values <0.001, Supplementary Dataset 4). In addition to the grouping factor, covariates including BMF score, sex, osmotic laxatives, and proton pump inhibitors (PPIs) uses exhibited strong associations with microbial composition, while age, statin, antidepressant, benzodiazepines, and PD-specific drugs had minimal influences on compositional changes (Fig. 5b).

Correlations between taxa abundance and covariates were tested by applying MaAsLin 2. The model included four stages (reference = control) and all covariates as fixed effects, and family id as the random effect. Besides, taxa abundance in early PD group was modeled separately to assess the response of individual taxa to PD-specific drugs. Significant associations ($q$ value <0.25, Supplementary Dataset 10) derived from MaAsLin 2 test were plotted in Fig. 5a. We found that the depletion of butyrate-producing bacteria (e.g., *Lachnospira*, *[Eubacterium]_ventriosum_group*, and *Butyricicoccus*) remained significant in RBD and early PD. Similarly, increased *Collinsella* and a notable decrease of *[Eubacterium]_ventriosum_group* were observed in RBD-FDR relative to the control. In terms of the covariates, the associations between BMF score and taxa abundance were extensive and resembled that of disease progression. In particular, a higher score of BMF (i.e., slower colon transit) was strongly correlated with a lower and higher abundance of *Butyricicoccus* ($\beta = -0.34$, $q$ value = 0.009) and *Oscillospiraceae_UCG-005* ($\beta = 0.32$, $q$ value = 0.003), respectively (Fig. 5a, c). With regard to the drug effect, the use of antidepressants, statin, and osmotic laxatives seemed to have great impact on differential taxa (Supplementary Datasets 10 and 11). In particular, we observed a strong co-occurrence of antidepressant usage and RBD/

## The effect of microbiota on prodromal PD was partially mediated via bowel movement frequency

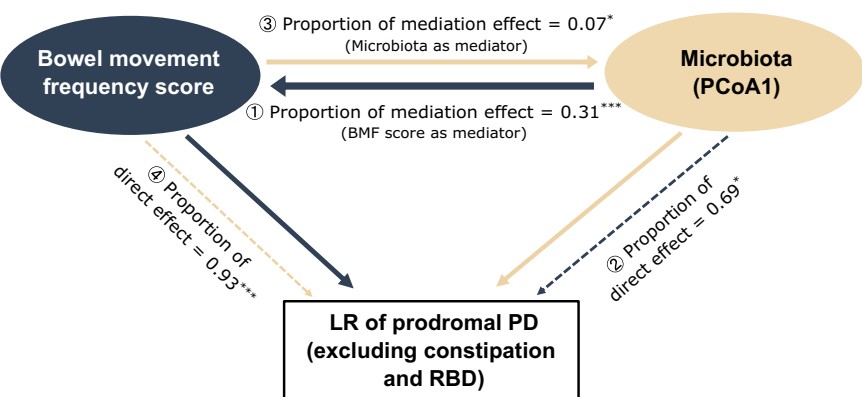

**Fig. 6 | Mediating effect of bowel movement frequency.** Generalized linear model mediation analyses were performed among control, RBD-FDR and RBD patients (*n* = 405), with microbiota (blue arrows) and BMF score (yellow arrows) as mediators, respectively. The proportion of mediation effect (①③, solid line) indicates the ratio of indirect effects (goes through the mediator) to the total effect, while direct effect (②④, dashed line) describes the proportion of exposure that directly affects the outcome after controlling for mediators. The significance of indirect/direct effects were assessed using bootstrapping procedures (two-sided test, *P* values were not adjusted). The effect of gut microbiota (exposure) on the likelihood ratio (LR) of prodromal PD (outcome) was partially mediated via bowel movement frequency (mediator) (proportion of mediation effect = 0.31, *P* = 0.0004). "***", "**", and "*" represented a *P* value less than 0.001, 0.01, and 0.05, respectively. RBD REM sleep behavior disorder, RBD-FDR first-degree relatives of patients with RBD, PD Parkinson's disease, BMF bowel movement frequency, PCoA1 first principal component from the principal coordinates analysis, LR likelihood ratio. Source data are provided as a Source Data file.

early PD-enriched genera (e.g., *Akkermansia* and *UBA1819*) in our data (Fig. 5c).

### Constipation (bowel movement frequency) mediates the effect of microbiota on α-synucleinopathy

Prior studies have suggested the bidirectional causal relationship between gut dysbiosis and constipation[30]. Given persistence of the distinct microbial changes in RBD and RBD-FDR even after adjusting bowel movement frequency, we further hypothesized that constipation may not entirely act as a confounder, but a mediating factor in the potential causal pathway linking gut microbiota and α-synucleinopathy. Thus, we conducted a mediation analysis in control, RBD-FDR and RBD patients (*n* = 405). The model was built by using total likelihood ratio of prodromal PD (excluding items of RBD and constipation) as the outcome, the value of first principal component (derived from the PCoA analysis) and BMF score as the exposure and mediator, respectively. We found that around 30% of the total effect of gut microbiota on prodromal PD went through the mediator (i.e., BMF score), indicating the potential direction of causality from gut dysbiosis, and constipation to α-synucleinopathy (Fig. 6).

### Changes of microbiota metabolism at prodromal and early α-synucleinopathy

Finally, we used PICRUSt2 (Phylogenetic Investigation of Communities by Reconstruction of Unobserved States) analysis to identify MetaCyc pathways that differed at prodromal and early stages of α-synucleinopathy. Kruskal–Wallis test showed that 18 metabolic pathways were significantly different among four groups (*q* value <0.05, Supplementary Dataset 12). Consistent with previous findings in PD patients[9], we observed that short-chain fatty acids metabolism (fermentation to lactate, ethanol, and acetate) and carbohydrate biosynthesis pathways were enriched, whereas cofactor and vitamin biosynthesis ($B_1$, $B_2$, and $B_{12}$) were decreased at prodromal and early α-synucleinopathy (i.e., RBD or early PD) (MaAsLin 2, *q* value <0.05, Supplementary Datasets 13 and 14).

Further multivariable association analysis showed that the enrichment of carbohydrate biosynthesis at prodromal and early α-synucleinopathy appears to be notably affected by BMF score, age, and

statin use. Similarly, the depletion of B vitamins synthesis pathways was strongly associated with sex, benzodiazepines, and osmotic laxatives uses (Fig. 7 and Supplementary Dataset 15). Nevertheless, altered microbial fatty acids metabolism (SCFA to lactate and ethanol) and preQ$_0$ biosynthesis (7-deazapurine biosynthesis) in RBD-FDR, RBD, and early PD remained significant relative to control even after adjusting all potential covariates. In addition, the salvage and de novo pathways of vitamin $B_{12}$ biosynthesis were significantly enriched in control than in RBD-FDR group.

## Discussion

Gut microbiota disturbances are well established in α-synucleinopathies, including Parkinson's disease. However, the emergence of microbial changes in the long prodromal period of PD is largely unclear. Here we investigated gut microbiota across early PD, RBD (prodromal PD), RBD-FDR, and control. We found that in RBD patients, the overall microbiota composition shifted closely to early PD, with depletion of butyrate-producing bacteria, and overabundance of *Collinsella*, *Desulfovibrio*, and *Oscillospiraceae UCG-005*. In RBD-FDR, an even earlier prodromal stage and younger population, there were emerging RBD/PD-like microbial changes, with regard to the increase of pro-inflammatory *Collinsella* and depletion of butyrate-producing *[Eubacterium]_ventriosum_group*. The predicted functional profile showed an overall increase in fatty acids fermentation to lactate and ethanol, and lower levels of deazapurine biosynthesis in RBD-FDR, RBD and early PD. Finally, we identified that host factors, especially bowel movement frequency (also act as a mediator), sex, age, and drug uses (e.g., antidepressant, statin, and osmotic laxative) could partially confound microbial changes in RBD-FDR, RBD and early PD. In summary, gut dysbiosis are already present at a much earlier stage, preceding the onset of RBD and PD, which emphasizes the potential role of gut microbiota in the pathogenesis of α-synucleinopathy.

Our findings suggest prominent microbial alterations (i.e., gut dysbiosis) at preclinical prodromal stages of PD. Consistent with prior reports in patients with v-PSG diagnosed RBD and possible RBD defined by screen questionnaire[21,23], we observed the shift of microbial community in RBD patients as compared with control. In addition, our data showed that the alteration of gut microbiota remained significant

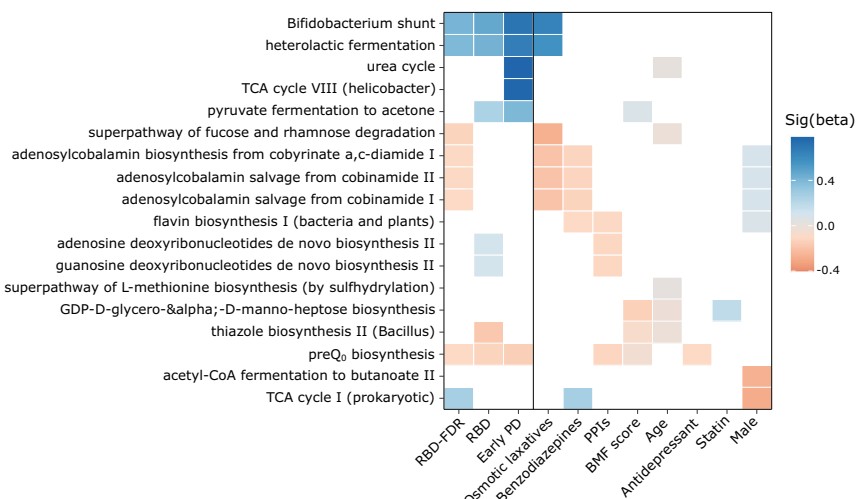

**Fig. 7 | Changes of microbiota metabolism at prodromal and early α-synucleinopathy.** The associations between pathway abundance and host factors were assessed by using MaAsLin 2. Only significant results were shown in the heatmap (Benjamini−Hochberg adjusted *P* values <0.25). Pathways related to short-chain fatty acids metabolism (e.g., Bifidobacterium shunt and heterolactic fermentation) and preQ$_0$ biosynthesis were consistently increased and decreased at prodromal and early stages of α-synucleinopathy even after adjusting all potential covariates, respectively. For detailed descriptions and results of MaAsLin 2, see Supplementary Dataset 15. RBD REM sleep behavior disorder, RBD-FDR first-degree relatives of patients with RBD, PD Parkinson's disease, MaAsLin 2 Microbiome multivariable associations with linear model, TCA tricarboxylic acid. Source data are provided as a Source Data file.

in RBD patients of disease duration less than 5 years as well as RBD patients who were supposed to have a lower risk of developing PD (i.e., probability of prodromal PD ≤ 80%) (Supplementary Fig. 8). From the perspective of individual taxa, we found that the hallmark of PD-like gut dysbiosis, namely the depletion of SCFA-producing bacteria (e.g., *Lachnospira* and *Butyricicoccus*), has already occurred at prodromal PD (RBD and RBD-FDR). This finding was not captured in prior studies probably due to the limited sample size of RBD patients[21,22]. In particular, a reduction of butyrate-producing *[Eubacterium]_ventriosum_group*[31] was already seen in RBD-FDR, which became more pronounced in RBD and early PD patients. Short-chain fatty acids, especially butyrate, are used by epithelial cells of colon as the source of energy and act as the modulator of tight junctions between adjacent epithelial cells[32]. Depletion of butyrate-producing bacteria may disrupt the integrity of intestinal barrier, and contribute to intestinal hyperpermeability, activation of enteric immune response and subsequent enteric α-syn aggregation[11,13]. In addition, *[Eubacterium]_ventriosum_group* has long been recognized as an anti-inflammatory bacterium that negatively correlates with plasma levels of interleukin-6 (IL-6), IL-8 and C-reactive protein[33,34]. These pro-inflammatory cytokines could enter CNS via blood-brain barrier, and possibly initiated the neuroinflammatory reactions and pathological processes of PD[35,36].

On the other hand, we observed that a group of gut microbes were consistently increased at prodromal and early stages of α-synucleinopathy. In addition to the genera *Akkermansia* and *Oscillospiraceae UCG-005*, which have been identified as RBD-enriched bacteria in a previous study[21], we also found that bacteria (e.g., *Collinsella* and *Desulfovibrio*) which were potentially related to PD pathogenesis were already increased in RBD patients[37,38]. In particular, *Desulfovibrio* is a hydrogen sulfide (H$_2$S) and lipopolysaccharide (LPS)-producing bacteria[37]. In the preclinical study, LPS-treated mice would present intestinal hyperpermeability and a higher level of pathological α-syn accumulation[13]. The production of H$_2$S also had a potential of inducing α-syn oligomerization[37], an initial step of α-synucleinopathy[39], and inhibiting gastrointestinal motility[40]. Interestingly, *Collinsella* is a hydrogen-reducing bacteria that could cross-feed with *Desulfovibrio*[41]. Enriched *Collinsella* was widely reported in low-fiber diets and

metabolic diseases (e.g., type 2 diabetes)[42,43]. One study in rheumatoid arthritis showed that *Collinsella* may lead to intestinal hyperpermeability by down-regulating the expression of epithelial tight junction[44]. Besides, *Collinsella* was associated with a higher level of pro-inflammatory IL-17A, which could exacerbate neuroinflammation especially microglial activation in PD rodent models[44,45]. The impact of *Oscillospiraceae UCG-005* (i.e., *Ruminococcaceae UCG-005*) on human diseases was largely unknown, albeit it steadily increased in Japanese and our RBD cohorts[21]. A recent study suggested the relationship between *Oscillospiraceae UCG-005* and low level of physical activity (a risk factor of PD)[46], albeit this correlation was not captured in our current study (Supplementary Fig. 9a). Finally, we observed an increase of mucin-degrader (i.e., potential gut barrier disrupter) *Akkermansia* in RBD and early PD, while the abundance of *Akkermansia* was significantly influenced by other covariates, such as antidepressant use. Previous study showed that mucin-degraders (family Prevotellaceae) were enriched in patients with depression and anxiety[47], and a further subgroup analysis of present study also supported a near-significant increase of *Akkermansia* in unaffected controls with antidepressant (n = 5, *P* value = 0.056) when comparing to those without (Supplementary Fig. 9b). Interestingly, there is a possibility that the disruption effect of *Akkermansia* and other mucin-degraders might be reversed by switching to a high-fiber diet, indicating the potential of dietary intervention in intervening gut dysbiosis at prodromal and early α-synucleinopathy, especially among patients with concomitant anxiety and depressive disorders[48].

In this study, the functional profile of gut microbiota was predicted from 16S rRNA gene sequencing data, suggesting increased fatty acids metabolism, and decreased biosynthesis of preQ$_0$ and vitamin B$_{12}$, at prodromal and early α-synucleinopathy. The enrichment of lactate production pathway is consistent with the increase of lactic acid bacteria (LAB) in PD patients[9]. However, the role of lactate in the pathogenesis of PD remains unclear[49], as it appears to conflict with the health-promoting effects of LAB. One possibility is that lactate might be further metabolized by other bacteria (e.g., non-butyrate-producing lactate-utilizing bacteria) and finally convert to the products that might be detrimental to gut health (e.g., disrupt gut

barrier)[50]. Similarly, decreased vitamin $B_{12}$ biosynthesis in PD patients have also been reported[9]. Low serum $B_{12}$ level is common in early PD and related to the neuropathy in PD patients[51]. However, $B_{12}$ deficiency in PD is likely not attributed to the altered gut metabolism, but related to a reduced dietary intake or digestive disorders[52]. Nonetheless, vitamin $B_{12}$ is an important modulator of gut microbial composition and metabolism, and it has been found that $B_{12}$ supplementation may boost the production of protective butyrate[52]. Finally, we found a consistent reduction in $preQ_0$ biosynthesis at prodromal and early α-synucleinopathy. $PreQ_0$ is a key intermediate of the biosynthesis of 7-Deazapurine nucleoside, the latter of which can form a variety of compounds with antibiotic and anti-cancer effects[53]. Although the role of 7-Deazapurine is still elusive, the abundance of $preQ_0$ biosynthesis seems strongly co-occurring with genus *Faecalibacterium*, a butyrate-producing bacteria and an indicator of PD progression (Fig. 3d and Supplementary Fig. 7).

In line with previous studies, our data showed that constipation symptom especially bowel movement frequency was a strong confounder on gut microbiota[23,54]. Nonetheless, majority of the gut microbial features identified at prodromal and early α-synucleinopathy remained robust even after adjusting BMF score, indicating that some features of gut dysbiosis are specific for α-synucleinopathy and emerge before the changes in bowel habits. Further mediation analysis suggested the potential causal pathway linking gut dysbiosis to constipation at prodromal α-synucleinopathy. This observation could be supported by prior clinical trials in Parkinson's disease, that PD patients treated with pro-/prebiotics had significantly increased spontaneous bowel movement[55]. Nevertheless, in current study, we found that over 50% of constipated RBD and early PD patients who were already receiving regular treatment of constipation still had decreased bowel movement (i.e., >every 3–4 days). Hence, interventions on constipation and specific microbes at prodromal stages of disease (e.g., RBD and RBD-FDR), may be a promising strategy for future prevention and disease-modifying therapy of α-synucleinopathy.

This study has some limitations. Firstly, this is a cross-sectional study, although it applied a staging concept to simulate the correlation of microbiota with the progression of α-synucleinopathy. Future prospective studies are warranted to verify the causal relationships between gut dysbiosis and α-syn pathology. Secondly, the sample size of early PD was relatively modest. Nonetheless, we still observed significant and consistent changes of gut microbiota in PD as similarly reported in previous studies[9]. Thirdly, this study focused on the gut-to-brain staging model of PD ("body-first"), therefore, additional research is needed before we can generalize the findings to the prodromal stages of other PD subtypes (e.g., "brain-first")[17]. Fourthly, RBD-FDR, which reflected an even earlier prodromal stage, were composed of younger and more female subjects. Nevertheless, the key findings remained robust even after adjusting for these sociodemographic differences. Also, the similar changes of gut microbes in RBD/early PD and their relatives (RBD-FDR) might be affected by other unmeasured factors, such as genetics, shared dietary habit and early-life exposure. Future analyses incorporating these factors may help understanding the development of gut dysbiosis at prodromal and early α-synucleinopathy. Finally, this study is limited to the compositional profile of gut microbiota and functional prediction based on 16S rRNA gene sequencing data, a comprehensive survey of gut microbiota (at species- and strain levels) and gut metabolism by metagenomics and metabolomics are needed in the subsequent studies.

In conclusion, our study suggested that PD-like gut dysbiosis occurs at the prodromal stages of PD, among patients with v-PSG-diagnosed RBD and their first-degree relatives. Future prospective studies, combined with investigations of gut metabolism, inflammatory markers, and enteric α-syn pathology, are required for a thorough understanding of the role of gut microbiota in the pathogenesis of α-synucleinopathy.

## Methods

### Subject recruitment
The study was approved by the Joint CUHK-NTEC Clinical Research Ethics Committee (CRE-2017.670) and registered at clinicaltrial.gov as NCT03645226. Written informed consent was obtained from all subjects in accordance with the Declaration of Helsinki. Figure 1 demonstrates the recruitment and flow of this study. Subject recruitment and stool sample collection were conducted in the Li Chiu Kong Family Sleep Assessment Unit, Department of Psychiatry, the Chinese University of Hong Kong (CUHK), between 2018 and 2021.

RBD patients and RBD-FDR were recruited from the ongoing RBD cohorts[15,20]. Controls were neurologically healthy subjects recruited from the community and our sleep clinic[15,20]. All RBD patients were diagnosed by video-polysomnography (v-PSG) according to the international classification of sleep disorders (3rd). A diagnosis of probable RBD was made using the Diagnostic Interview for Sleep Patterns and Disorders (DISP)[25], to screen control and RBD-FDR subjects who likely had RBD or isolated RBD features in the absence of v-PSG examination. To obtain a more clearly differentiated staging model, we excluded RBD-FDR with v-PSG-confirmed RBD and neuro-degenerative diseases[20]. Likewise, controls with neurodegenerative diseases, probable RBD, v-PSG diagnosed RBD or a positive family history of RBD symptoms were excluded from the staging model. In this study, we used the term control to represent these neurologically healthy subjects.

Patients with early PD were recruited from (1) PD ($n = 17$) converted from our RBD cohorts[15], and (2) PD patients with a preceding history of RBD as referred by neurologists ($n = 19$). Inclusion criteria for early PD group including: (1) a diagnosis of PD confirmed by the neurologist with reference to the standard diagnostic criteria[56]; (2) parkinsonism less than 5 years, (3) with no dementia (i.e., a total score of Hong Kong version of Montreal Cognitive Assessment [HK-MoCA] > 21 or Clinical Dementia Rating [CDR] < 1); (4) had v-PSG diagnosed RBD; and (5) parkinsonism preceded by RBD features.

Other exclusion criteria were as follows: (1) RBD-FDR in cohabitation with RBD patients; (2) antibiotics usage within one month; (3) pre-existing gastrointestinal diseases that prominently confound gut microbiota, e.g., inflammatory bowel diseases and liver cirrhosis (Supplementary Fig. 1).

### Questionnaire and clinical assessment
General questionnaire consisted of sociodemographic features, life-style (e.g., smoking, coffee drinking, and exercise), excessive daytime sleepiness (Epworth sleepiness scale, ESS), autonomic function (Scales for Outcomes in Parkinson's Disease−Autonomic, SCOPA-AUT). Biological sex was based on self-report, and the information was further verified from the clinical management system of hospitals in Hong Kong. Besides, we assessed the severity of RBD features among all subjects by using the RBD questionnaire Hong Kong (RBDQ-HK), which consists of two components: (1) dream-related factors (factor 1) and (2) behavioral manifestations (factor 2)[57]. Bowel disorders, such as functional constipation, irritable bowel syndrome, and functional diarrhea, were diagnosed according to the Rome-IV diagnostic questionnaire for adults[27]. In addition, we documented stool consistency (BSFS, range from 1 to 7) and bowel movement frequency score (Q: "How often do you have a bowel movement in past three months?" 1 = ">1/day", 2 = "1/ day", 3 = "every other day", 4 = "every 3–4 days", 5 = "every 5–6 days", 6 = "≤1/week"). Upper gastrointestinal symptoms, including dysphagia (e.g., swallowing/choking), sialorrhea, and early satiety were derived from SCOPA-AUT.

Parkinsonism was calculated by using the Unified Parkinson's Disease Rating Scale part III (UPDRS-III)[58], possible subthreshold parkinsonism was defined as the total score of UPDRS-III (excluding action tremor) larger than 3[59]. Orthostatic hypotension was defined by the reduction of systolic blood pressure (SBP) ≥ 20 mmHg or diastolic

blood pressure (DBP) ≥ 10 mmHg at stand position[60]. If subjects had supine hypertension (SBP ≥ 140 mmHg or DBP ≥ 90 mmHg), then SBP drop ≥30 mmHg or DBP drop ≥15 mmHg was considered as orthostatic hypotension. Olfactory function was assessed by using the Olfactory Identification Test (OIT)[61]. The total correct score of OIT less than 3 was considered as olfactory impairment[20]. Psychiatric disorders were interviewed by psychiatrists with MINI International Neuropsychiatric Interview (M.I.N.I.)[62]. We applied Hong Kong version of Montreal Cognitive Assessment (HK-MoCA) to estimate cognitive function[63]. Subjects with a total score of HK-MoCA ≤21 was considered as having global cognitive deficit[63]. Furthermore, we calculated the total likelihood ratio (LR) and probability of prodromal PD according to the updated Movement Disorder Society (MDS) research criteria (2019)[64]. A probability of prodromal PD > 80% and 30–80% were considered as probable and possible prodromal PD, respectively.

## Stool sample collection and 16S V3-V4 rRNA data processing

The fresh stool samples were collected on the day of clinical interview, during the overnight sleep assessment or at home. On the day of collection, samples were stored in sterile containers, and transferred to the laboratory in a stool kit (containing ice gel, polystyrene box, and thermal bag) within four hours. Meanwhile, subjects reported the information, including collection time, BSFS, probiotics and medications taken within 3 days. When samples arrived laboratory, they were aliquoted into 1–2 vials and stored in −80 °C freezer for future analysis.

DNA extraction was performed using the DNeasy PowerSoil Pro DNA Kit (Cat. No.: 47014, Qiagen). The concentrations of the extracts were measured by NanoDrop 2000 spectrophotometer (Thermo Fisher Scientific). The DNA library was constructed using primers spanning target hypervariable regions V3-4 of the 16S ribosomal RNA genes (341 F: 5′-CCT ACG GGN GGC WGC AG-3′, 806 R: 5′-GGA CTA CNV GGG TWT CTA AT-3′), together with adapter sequences and unique 12 bp barcodes indexed to the forward and reverse primers. PCR amplicon was sequenced on an Illumina MiSeq platform using paired-end 300 bp reads. Sequencing reads were denoised into amplicon sequence variants (ASVs) using DADA2 (q2-dada2 plugin)[65] in QIIME2 software (v2021.4). Forward and reverse reads were truncated at position 288 and 272, respectively, where there was a significant drop in Phred quality score. Samples with low total frequency (i.e., total read count <1000) were filtered via q2-feature-table. Taxonomy was assigned to ASVs using the q2-feature-classifier[66] classify-sklearn naïve Bayes taxonomy classifier against the SILVA v138 99% 16 S rRNA databases[67]. The taxonomy table was then collapsed at levels of genus, family and phylum, and the merged abundance table was used for downstream analysis.

## Clinical characteristics analysis

The data were assessed for normal distribution before performing statistical analysis by the Shapiro–Wilk test. The univariate analysis of categorical data would be performed by the Chi-square test or Fisher's exact test, where applicable. For continuous data with normal distribution, ANOVA followed by a post hoc test was used, otherwise, the Kruskal–Wallis H test was used. Considering the potential association between subjects from the same family, we applied Generalized Estimation Equation (GEE) model to adjust family clustering in the comparisons of gastrointestinal features, clinical characteristics, and neurodegenerative markers among groups[20]. GEE model was constructed with family id as the cluster factor and subject id as the within-cluster factor, using an independent correlation structure. The type of model was specified as Linear (continuous variables) or Binary logistic (binary variables) based on the distribution of the dependent variables. Tests were performed using IBM SPSS Statistics for Windows, Version 26.0 (Armonk, NY: IBM Corp). A two-sided $P$ value less than 0.05 was considered as statistically significant. For post hoc multiple comparisons in the GEE model (i.e., between each pair of groups),

Benjamini–Hochberg False Discovery Rate was used to adjust $P$ values, and a false positive rate less than 5% (i.e., $q$ value <0.05) was accepted and indicated statistical significance.

## Microbiota community composition

Alpha diversity (Chao 1, Gini Simpson, and Shannon indexes) of each sample was assessed by using vegan R package (version 2.6-2) with R project Version 4.2.1., and statistical significances were tested using Kruskal–Wallis (KW) test with post hoc analyses. Compositional differences between each pair of groups were analyzed using permutational multivariate analysis of variance (PERMANOVA, 99,999 permutations) in the vegan R package ("adonis2" function), and distance matrix for "adonis2" function was constructed based on the Bray–Curtis distance of the relative abundance of whole genera ("vegdist" function). Besides, we used multivariate homogeneity of groups dispersions (multivariate extensions of Levene's test) to examine the homogeneity of variances between groups ("betadisper" function, vegan). Compositional shift at prodromal and early α-synucleinopathy was further visualized using principal coordinates analysis based on the same distance matrix ("wcmdscale" function from R package vegan).

## Differential abundance analysis

The taxonomy table was filtered for genera and family with low abundance (<0.05%) and/or low prevalence (<10%) in QIIME2 (via qiime feature-table filter-features-conditionally), and 88 out of 249 genera as well as 36 out of 84 family were retained for differential abundance analysis. In pre-processing, read count table was reframed to compositional data with centered log-ratio (CLR) transformation (after applying a pseudocount of 1) using clr function from the compositions R package (version 2.0-4). To test the concordance of the changes in taxa abundance from control, RBD-FDR, RBD to early PD, we used the nonparametric Kendall's tau-b correlation coefficient implemented in SPSS (version 26.0). Differential genera between groups were performed using Kruskal–Wallis test (SPSS 26.0). The correlations of significantly differed taxa ($q$ value threshold 0.05, Kruskal–Wallis test) and different stages of α-synucleinopathy were further validated by applying Microbiome multivariable associations with linear model (MaAsLin 2)[68] ("Maaslin2" function within the MaAsLin 2 R package, version 1.8.0). The model included family id as the random effect to incorporate variability in family clustering (e.g., RBD-FDRs from the same family). Multiple comparisons were adjusted using Benjamini–Hochberg method. Considering the input data were CLR-transformed abundance, we turned off the default normalization and transform methods implemented in MaAslin2 function.

## Random forest classification

Cross-validation random forest machine learning algorithm was performed to estimate the accuracy of gut microbiota in discriminating RBD from control and RBD-FDR group. The predictors were CLR-transformed abundance of filtered genera ($n = 88$) as described in the above differential abundance analysis. For each prediction model, the dataset (e.g., control and RBD) was divided into a training set (80% of whole dataset) and a test set (20%). In order to reduce sampling error, the original case/control ratios were preserved in the new datasets via stratified sampling. We applied recursive feature elimination (RFE) algorithm to select predictors from the training set, using the "rfe" function (25 repeats of tenfold cross-validation) from caret R package (version 6.0-92). The final list of predictors kept in trained model were determined by the prediction accuracy, and the performance of the model was further evaluated in the hold-out test set. The whole process from data splitting, feature selection/model fitting, to prediction evaluation was repeated 25 times (i.e., resample 25 times, Supplementary Fig. 5). Receiver operating characteristic (ROC) curves and the area under the curve (AUC) were calculated with R package "pROC" (version 1.18.0).

### Analysis of host−microbiome interactions

Host factors that exhibited significant or near-significant differences between groups were considered potential confounding factors. The association of host factor and global microbial composition was calculated by PERMANOVA based on the Bray−Curtis distances of all genera (permutation = 99,999). Besides, we used MaAsLin 2 to elucidate host factors contributions to differential taxa identified at prodromal and early α-synucleinopathy, therefore, this model was trained with four stages and all covariates as the fixed effects, and family id as random effect. Benjamini−Hochberg method was used to adjust $P$ values generated by multiple comparisons from PERMANOVA and MaAsLin 2 tests, where the false discovery rate threshold for MaAsLin 2 was 0.25[68].

### Mediation analysis

Statistical significance of mediation effect was analyzed using R package "mediation" (version 4.5.0). To improve the power of analysis, we used the total LR of prodromal PD (i.e., the risk of developing prodromal PD) as outcome, and either microbiota (first principal component) and constipation (represented by bowel movement frequency score) as the mediator or exposure. The average causal mediation effects (ACME) indicated the indirect effect that goes through the mediator, while average direct effects (ADE) described the direct effect of exposure on the outcome. We tested the significance of this indirect/direct effect using bootstrapping (10,000 resamples) procedures. The proportion of the effect of the exposure on the outcome that goes through the mediator was evaluated by dividing the ACME by the total effect (ACME + ADE).

### Prediction of functions from 16S marker sequences

Finally, we used PICRUSt2 (Phylogenetic Investigation of Communities by Reconstruction of Unobserved States) software (https://github.com/gavinmdouglas/q2-picrust2/releases/tag/2021.11_0)[69] to predict the functional potential of microbial community. Metabolic pathways were annotated using the MetaCyc database[70]. Similarly, we filtered out low prevalence (<10%) and abundance (<0.05%) pathways and transformed abundance data with CLR method for further analysis. Differential abundance of pathways was identified by using Kruskal−Wallis test, and the correlations of differential pathways with different stages of α-synucleinopathy were assessed by applying MaAsLin 2, as elaborated in the above differential taxa abundance analysis.

### Statistics and reproducibility

Sample size for this study was predetermined in research proposals funded by Health and Medical Research Fund of the Food and Health Bureau (Ref No.: 05162876, 2017) and Center for Gut Microbiota Research, Faculty of Medicine, the Chinese University of Hong Kong. The calculated effect size (Cohen's $d$) for the differences in microbiota abundance between PD and control was 2.68. It was expected that the effect size for the differences among patients with RBD, RBD-FDR, and control was smaller, and therefore, an effect size with a Cohen's $d$ of 0.80 was employed in the sample size estimation. Based on this figure, at least a sample size of 36 in each group was required to achieve a type I error of 0.05 and a power of 0.90 in 4 groups analysis of variance (ANOVA), with consideration of a numerator degree of freedom of 6. For the data collection, all investigators were blind to group assignment at clinical interview and biological sample processing. As mentioned above, a total of 11 samples were excluded for subsequent analyses due to the low quality of sequencing data, even after repeating DNA extraction, amplification and sequencing. In addition, to further improve the reproducibility and replicability of this study, we applied: (1) different statistical analyses methods (e.g., Mann−Kendall trend, Kruskal−Wallis tests and MaAsLin 2) to verify specific research question, i.e., whether gut microbiota have changed at prodromal and early stages of α-synucleinopathy; (2) a nested cross-validation and repeating procedures in machine learning classifier to calculate the average performance of microbial markers in differentiating RBD from control and RBD-FDR. The seeds set for train-test splits and random forest models in R project were provided at GitHub repository; (3) for multiple comparisons, Benjamini−Hochberg false discovery rate were used to further control false positive findings.

## Data availability

All sequencing reads generated in this study have been deposited in the European Nucleotide Archive (ENA) at EMBL-EBI under accession number PRJEB52086. All processed sequencing data, sociodemographic and clinical characteristics data generated or analyzed during this study are available at GitHub repository (https://github.com/Joannehb/Gut-microbiota-across-early-synucleinopathy). The Silva 16S rRNA database used for alignment is available at https://data.qiime2.org/2020.6/common/silva-138-99-nb-classifier.qza. Source data are provided with this paper.

## Code availability

Custom code used for processing and analyzing the data in this study have been deposited in the Zenodo repository (https://doi.org/10.5281/zenodo.7783875) and is also available at GitHub (https://github.com/Joannehb/Gut-microbiota-across-early-synucleinopathy).

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

## Acknowledgements

We would like to thank all the subjects who have participated in this study. This study was funded by the Health and Medical Research Fund of the Food and Health Bureau (Ref No.: 05162876 to Y.K.W.) and Research Grants Council (RGC-CRF Ref No.: C4044-21GF to Y.K.W.) of Hong Kong, China, and Center for Gut Microbiota Research, Faculty of Medicine, The Chinese University of Hong Kong (to Y.K.W. and S.H.W.). B.H. and J.W. were supported by the Faculty Postdoctoral Fellowship Scheme, Faculty of Medicine, CUHK.

## Author contributions

Y.K.W., J.Z., Y.L., P.K.S.C., S.H.W., V.C.T.M., K.F.T., S.N., and F.K.L.C. contributed to the conception and design of the study; B.H., S.W.H.C., Y.L., J.W.Y.C., J.W., S.L.M., Y.K.Y., Z.C., H.M.L., L.Z., and Y.K.W. took responsibility for the integrity of the data and preparation of the manuscript; B.H., J.Z., S.W.H.C., J.W.Y.C., Y.K.Y., C.T., and Y.K.W. contributed to drafting the text and preparing the figures.

## Competing interests

Y.K.W. received personal fees from Eisai Co., Ltd for lecture, travel support from Lundbeck HK Limited and J.W.Y.C. received personal fees for joining an expert panel meeting of Eisai Co., Ltd., which are outside the submitted work. The Chinese University of Hong Kong has filed a U.S. provisional patent (application no. 63/446,304, filed on February 16, 2023) relating to the use of microbial markers for the early detection of prodromal PD (RBD and RBD-FDR), on which Y.K.W., H.M.L., B.H., P.K.S.C., V.C.T.M., and F.K.L.C. are inventors. B.H. and J.W. were supported by the Faculty Postdoctoral Fellowship Scheme, Faculty of Medicine, CUHK. The remaining authors declare no competing interests.
