## [Peer Review File · Nature Communications]

Gut microbiome dysbiosis across early Parkinson's disease, REM sleep behavior disorder and their first-degree relativesReviewers' comments:

Reviewer #1 (Remarks to the Author):

This study reports the key finding that alterations in gut microbiome with depletion of short-chain FA-producing bacteria (SCFA) and overabundance of Akkermansia/other bacteria inducing gut barrier disruption are seen not only in PD, but also in polysomnographically diagnosed RBD patients and their first degree relatives. This raises the potential that early interventions targeted at gut microbiota in RBD and first-degree RBD relatives might slow or prevent progression from RBD to PD.

Only 3 previous studies have reported the gut microbiome in RBD- two using PSG-diagnosed RBD cohorts as referenced by the authors in ref 20-21, however a third study from the TREND German cohort found individuals with possible RBD diagnosed on screening questionnaire showed altered beta-diversity in microbial composition. DOI: 10.1002/ana.26128. I feel this should be added to the Discussion for completeness.

Strengths of this study include the largest numbers of RBD and controls published to date (n=170 and 108 respectively), hitherto unpublished data on RBD first degree relatives (n=127) with 36 early PD subjects. A variety of appropriate statistical approaches have been adopted for this complex subject, including GEE modelling, univariate analysis, PERMANOVA, PCOA and RF modelling showing a progressive decrease in SCFA and increased Akkermansia and related species from control to RBD-FDR, to RBD to PD.

I have a few comments and queries:

1. Little detail is given in the methods about patient selection- how was early PD defined, ie were participants within 3 years of PD diagnosis? The RBD-FDR group are defined as "high-risk" yet Figure 1 suggests they were sequentially recruited, and excluded if they had PD, cirrhosis or colonic cancer. The authors should stipulate that no further stratification or exclusion was performed on the RBD-FDR group. Also, the diagnosis of probable and definite RBD should be defined in the methods and not just in Figure 1, unless I have missed this. What was the split between probable and definite RBD cases, and the rationale to lump them together in the analysis? Was a microbiome gradient seen from possible to probable RBD in the data?

2. How generalizable are the author's findings to brain-first PD, that is in PD patients whose PD was not preceded by clinical or PSG-defined RBD? This is a potential limitation of the study, in that only half of the 36 early PD patients are in this category, and this should be discussed in the Discussions limitation section. This will be important in directing future therapies for PD patients.

3. Did the authors look at parallel measures of G.I mucosal inflammation or integrity from the faecal samples? This would be very interesting, to directly link changes in the microbiome to mucosal breakdown

4. The manuscript is well written, but would benefit from a general English review as I identified a number of minor language and figure errors:

Abstract: suggest change to: "There has been ongoing debate about the causal relationship between gut dysbiosis and Parkinson's disease (PD), partially due to the high variability of gut microbiota (sensitive to host lifestyle and physiological characteristics), and the difficulties in identifying subjects with a prodromal PD syndrome"

Introduction para 2: change language and insert definition of earlier or prodromal RBD:

"Patients with isolated/idiopathic RBD reported an increased prevalence of constipation, and show increased phosphorylated α -syn immunostaining in ENS. Likewise, PD patients with premotor RBD features appeared to exhibit prominent degeneration of peripheral nervous system (e.g., increased constipation and enteric α -syn histopathology) when compared to those without, suggesting that PD with RBD is a distinct subtype of Parkinson's disease reflecting the gut-brain hypothesis of α -synucleinopathy. On the other hand, isolated RBD features, e.g., isolated dream-enactment behaviors not meeting RBD polysomnographic diagnostic criteria, might reflect a prodromal stage of RBD and the early presentation of α -synucleinopathy..."

Intro: para 3, last sentence- suggest reference here TREND study which showed that individuals with possible RBD identified on screening questionnaire showed altered beta-diversity in microbial composition. DOI: 10.1002/ana.26128

Figure 4-Differential gut microbes at the early stages of α -synucleinopathy-The use of 2 similar blue colors to denote RBD FDR and RBD means they are very difficult to distinguish in Figure 4. Suggest change

Discussion para 3, change to:

"Interestingly, a previous study showed that this disruption effect of Akkermansia and other mucin-degraders could be reversed by switching to a high-fiber diet, indicating the potential of diet therapy in intervening gut dysbiosis at early α -synucleinopathy" to minimise confusion

Discussion- last para, does not read well. Suggest consider changing to below or similar:

"In conclusion, our study suggested that gut microbiota might be a promising target for the early prevention of PD in those with polysomnographically diagnosed RBD and their first degree relatives who manifest earlier prodromal forms of RBD. Future prospective studies, combined with investigations of gut metabolism, inflammatory markers and enteric α -syn pathology are required for a thorough understanding of the role of gut microbiota in the pathogenesis of α -synucleinopathy. "

References:

Ref 33 appears incorrect or incomplete

Reviewer #2 (Remarks to the Author):

The article from Dr Bey Huang et al describes the results of a study aimed at characterizing the gut microbiota in the early stage of α -synucleinopathies. They performed a cross-sectional study in which they collected and analysed stool samples of controls, first-degree relatives of RBD patients, RBD patients, and early PD patients. They show that the alterations in gut microbiota composition observed in PD, especially the depletion of SCFA producers, appear already in RBD patients, suggesting a causal link between gut microbiota and PD.

First, I would like to congratulate the author on this really nice and novel piece of work. The study in my opinion is well planned, the overall recruitment and sampling strategies are well described and depicted. I think that overall, the analyses are well performed, and I particularly appreciate the creativity of the authors in presenting the data.

I think the paper deserves publication, but I have some concerns that I would appreciate be addressed before final acceptance.

General comments:

It is not clear to me how the authors analysed the 16S data. Qiime2 is a sort of wrapper program that uses other programs to analyse the data. Were OTUs or ASV called? How was this done? Which program was used? For example, "Sequencing tags and contaminant reads were removed by alignment tools, samples with low total frequency", what alignment tools? Name them, with version and reference.

The statistical description of the approaches used for analysing the microbiome can be greatly improved. Please, provide the details of the model structures, report versions of the packages, and include references to them. For example, explain why you choose to use GEE instead of LM. Also provide detail of how the GEE models were run, what type of data/correlation structure you used within the model. The authors declare in the reporting summary that no custom code was created. I am afraid I disagree. The authors had to manipulate their data, filter them, fit models based on their experimental design, extract estimates, errors, and p-value all tailored to their dataset. In my opinion, this is custom code. I believe the authors need to deposit their code in a public

repository to foster transparency and reproducibility. Similarly, the authors have to provide a link to the raw reads deposited in NCBI/ENA. I believe this should be done at this stage, as the data can be made private and reviewer access can be requested.

A less important point, more strength to the findings could be given by citing additional recent meta-analyses that reported depletion of SCFA producers in PD and enrichment of Akkermansia and Christensenellaceae (e.g. Shen et al 2021, Romano et al 2021, Toh et al 2022).

Detailed comments:

The DMM model used to identify the community types is not convincing for me. Similar approaches have been used to identify enterotypes and have been highly debated. In my opinion and experience, these approaches will provide a set of more or less robust clusters even though they have no biological meaning or are not significantly different. For example, in Figure 3A from the PCoA plot, I can see that in terms of dissimilarity distances the Type B and C are basically identical. D and E might be slightly different but also largely overlapping. Can please the authors provide additional evidence that the clustering is robust and meaningful. Similarly, they report in the Discussion that "The community type D resembled an unstable community type in Human Microbiome Project (HMP) 23 (supplementary table 10)." This seems to be based on checking what bacterial taxa (phyla) are mainly present in the description of these groups. Can the author provide any evidence that these are actually similar communities? I am afraid that a comparison just based on the cluster broad description is not enough. For example, in type D Faecalibacterium is low in abundance. However, it is also a Firmicutes, which is a group of bacteria enriched in the Type B community of Ding et al.

The authors mention a "pairwise comparison for the global composition of the gut microbiome" in pg 9 and afterward. How was this done? The permanova analysis, presumably done using the adonis2 function in vegan (as this was not specified) is not a pairwise test. There is no mention of this in the method section.

Also, in my opinion, Fig 2 is not well presented. I would first present an overall PCoA with all samples and report the pairwise comparison in a table. I would remove figure 2B. Fig 2A is very nice, and I will present it as Fig 2B. Just a minor note here, the authors mention that they tested for homogeneity of variance in the groups, however from the PCoA seems the Early PD and RBD_hr have much broader dispersion of points. Might this also explain why there is only a marginal difference between RBD_hr and RBD_lr? Also, where is the comparison between RBD_hr and PD?

I believe that the authors should do a little more effort in explaining how they performed the differential abundance analyses (DA). ANCOM accounts for the compositionality of the data. However, GEE will not. Were the data rarefied and transformed? How was this done? Also, why did they choose a cutoff for the ANCOM of 0.6? As a minor comment, ANCOM might have a high false-negative rate, especially in the first version the author used. So, even if not critical, the authors might consider using an updated versions or additional methods.

The data presented in Fig 5 and 6 are the ones that concern me the most. As far as I could understand the ML was built using the differential abundant taxa identified in the previous ANCOM analyses. However, these taxa have been selected on the whole dataset. This means that they will be selected on both the training and test dataset of the ML model. In my opinion this will generate some "data leakage" and inflate the AUCs. Can please the author clarify why this should be acceptable and reperform the RF model (if needed) on the whole dataset or at least perform variable selection only on the training set?

I do not fully understand fig 6A. Was this done using a permanova (as I can read in pg 29)? Where are the p-values? Permanova on the whole dataset will not report R² for each group. Please clarify.

Similarly, they write that they stratified the data by various confounding factors and then re-assessed whether SCFA taxa remained DA across groups, The conclusion was that "In general, the observed microbial changes as calculated by PERMANOVA test, GEE model and ROC analyses

remained robust in RBD patients after further stratification of confounders". However, it seems to me that this is not always the case and that sometimes an interaction is actually present in the data. What happens if the authors test whether the interaction between RBD:host-factors is actually defining taxa abundances? How should this be interpreted? I have the feeling that the real picture is more complex than what is presented. I would have liked to see also that the authors accounted for the confounding effect of medication on the SCF taxa abundance. It seems to me that RBD and PD took more antidepressants than the rest of the participants. Is this drug confounding taxa abundances? Also, it seems to me that in Fig 6 only data for RBD are shown. However, they mentioned that they did these analyses for RBD-FDR as well. Where are these data? The supplementary table would be clearer if they showed the results before and after stratification side by side. Finally, how "hard stool" was defined? Using the Bristol chart? Was this included as a binary variable in the model? Wouldn't be more appropriate to use the Bristol scores, or even better the water content of the stool?

I liked the mediation analysis very much. I think it would be very helpful if the authors could explain this procedure better to make it more accessible to the non-experts.

Conclusions:

The paper is of high quality.

I believe it cannot be published in this form.

Thorough adjustment of the method description, verification of the DMM, ML, and confounding analyses, and deposition of raw reads and R code in public repositories are needed.

Reviewer #3 (Remarks to the Author):

This is a timely and novel study of the potential changes in the fecal microbiome of subjects who are otherwise healthy controls (n=108 in final data set), or have a diagnosis of REM sleep behavior disorder (RBD; n=170), are first degree relatives of a subject with RBD (RBD-FDR; n=127), or have early-stage Parkinson's disease with a pre-motor symptom phase history of RBD-like symptoms (n=36). The study uses standardized clinical measures and instruments to ascertain and assess the subjects, and standard methods for performing 16S-based metagenomic sequencing analysis. They report a number of interesting findings, such as overall shifts in the microbiome composition in subjects with RBD who are at-risk for PD, and subjects with early stage PD compared to controls and RBD-FDR groups. Moreover, they also were able to identify 5 phylotypes based on shifts in composition determined by Dirichlet multinomial mixture modeling and observed that the subject groups differed in the phylotype they were most strongly associated with. For example, type D (associated with lower *Bacteroides*, *Faecalibacterium*, and higher *Bifidobacterium*) was more common and types A and B less common in RBD and early stage PD compared to controls and RBD-FDR. Further analysis by the authors revealed two specific short-chain fatty acid (SCFA) producing taxa - *Rosburia* and *Lachnospiraceae* - were reduced in all groups compared to controls, while one mucin-degrading bacteria was present at higher abundance in all groups compared to controls. The authors went on to assess the extent to which a diagnostic classifier could be developed based on the data for these and other studies, and claim to have achieved an ROC of 0.80 for distinguishing RBD subjects from controls, when using a classifier with 15 genera and the likelihood ratio for PD (exclusive of RBD-specific questions). Finally, the authors report some interesting associations between host activities and microbiome shifts after controlling for a number of potential confounds that they identified.

Overall, the study provides some convincing findings that would be of interest to many in the movement disorder field. However, there are a number of significant concerns that reduce overall enthusiasm. First, as a clinical study, although the ascertainment methods appear sound, the

composition of the subject groups is markedly different in many ways that could impact study outcomes considerably in a non-specific manner. For example, the RBD-FDR group is approximately 10 years younger than all other groups and notably more female. The RBD group also has twice as high a rate of PPI (proton pump inhibitor) and antidepressant use as well as statin use than the other groups, and the PD group has much higher use of dopamine-acting drugs. All of these things are not themselves part of the core pathogenetic mechanisms in RBD or PD, but they all are known to impact microbiome composition and gut function. Thus, it is hard to accept that some of the differences observed are not due to such non-specific influences on their data. Second, as a microbiome study, the methods and depth of analysis is somewhat superficial. It would have been very helpful for the authors to perform more sophisticated metabolic pathway profiling based on the compositional shifts seen between their groups, rather than make somewhat sweeping statements based on a select few number of taxa. Third, as a machine-learning diagnostic classification study, the work is simply not valid enough to draw any strong conclusions from. When evaluating the performance of a machine-learning tool, the use of 4-fold cross validation should not be viewed as an independent test of the performance of the model, but rather just a validation that an optimized model was developed. A true hold out set of fully-independent samples that were never used to help train the model must be used to establish its performance. There are clear standards of practice for the adoption of such approaches in precision medicine. Finally, from a statistical power point of view, the study simply has too few subjects to conclude that the results would be robust and reproducible in other cohorts. While RBD subjects and their relatives are somewhat difficult to find and recruit into studies, there are ample subjects of subjects with early stage PD.

Other minor concerns that merit correction include confirming whether the UPDRS was used instead of the UK Brain Bank criteria for verification of PD disease status in living subjects (page 25), whether any correction for multiple testing was performed on individual taxa data, or simply the nominal significance reported throughout, and inclusion of sequencing and quality assurance criteria parameters, such as the percentage of aligned reads, average coverage, length of trimmed reads, etc. One would like assurances that differences were not influenced by inherent biases in the raw sample material.

Comments and Responses

We greatly appreciated the insightful, helpful, and encouraging comments and suggestions of the reviewers. We have addressed all substantial concerns with regard to our study design, analysis, and data interpretation in the point-to-point responses.

Here we will first summarize the key revisions:

- 1) We have substantially improved the statistical description of the approaches used for analysing the microbiome in the revised section of Online Methods. The processing of 16S V3-V4 rRNA data was described in a clearer fashion. Details of data transformation, model structure, and package (or function) used in the analyses were provided.
- 2) The custom codes generated in this study have been uploaded to GitHub (<https://github.com/Joannehb/Gut-microbiota-across-early-synucleinopathy>), and raw sequence data have been deposited in the European Nucleotide Archive under project number PRJEB52086 (made public in July 2022).
- 3) In this revision, we used a more standard approach for random forest machine learning classification. The whole dataset was divided into a training (70%) and test set (30%), and a recursive feature elimination algorithm was employed to make feature selection from all filtered genera. The performance of the trained model was subsequently examined in the test set and displayed on a receiver operating characteristic curve.
- 4) The Dirichlet multinomial mixtures (DMM)-based method to identify microbial community component was removed from this study due to the unclear biological meanings of different community types as aptly suggested by the reviewer.
- 5) Differential abundance (DA) analysis methods were changed to Kruskal-Wallis test and MaAsLin 2 (Microbiome multivariable associations with linear model), with the aim of balancing the false discovery and false negative rates in the DA analysis.
- 6) We used PERMANOVA (permutational multivariate analysis of variance) and MaAsLin 2 to further delineate the effect of host factors (e.g., clinical profile, drugs) on the overall microbial composition and taxa/pathway abundance.

- 7) We have added the functional prediction of 16S marker sequences to enhance the recognition of microbial metabolic changes at early α -synucleinopathy.
- 8) The details of subject recruitment, including subject selection and inclusion/exclusion criteria, were explicitly described in the Online Methods and summarized in the Supplementary Fig.1.
- 9) All statistical results reported in the main text could be found in the corresponding supplementary tables and figures.

The more detail and relevant specific comments and suggestions from reviewers (**in bold**) and our responses/changes are as follows:

Reviewer #1 (Remarks to the Author):

Comment 1 : This study reports the key finding that alterations in gut microbiome with depletion of short-chain FA-producing bacteria (SCFA) and overabundance of Akkermansia/other bacteria inducing gut barrier disruption are seen not only in PD, but also in polysomnographically diagnosed RBD patients and their first degree relatives. This raises the potential that early interventions targeted at gut microbiota in RBD and first-degree RBD relatives might slow or prevent progression from RBD to PD.

Strengths of this study include the largest numbers of RBD and controls published to date (n=170 and 108 respectively), hitherto unpublished data on RBD first degree relatives (n=127) with 36 early PD subjects. A variety of appropriate statistical approaches have been adopted for this complex subject, including GEE modelling, univariate analysis, PERMANOVA, PCOA and RF modelling showing a progressive decrease in SCFA and increased Akkermansia and related species from control to RBD-FDR, to RBD to PD.

Response 1: We thank the reviewer in supporting the potential interest of our findings. We responded to his/her concerns point-by-point as follows.

Comment 2: Only 3 previous studies have reported the gut microbiome in RBD- two using PSG-diagnosed RBD cohorts as referenced by the authors in ref 20-21, however a third study from the TREND German cohort found individuals with possible RBD diagnosed

on screening questionnaire showed altered beta-diversity in microbial composition. DOI: 10.1002/ana.26128. I feel this should be added to the Discussion for completeness.

Response 2: Thanks for reminding us that we missed an important piece of work when interpreting our findings. The paper from TREND German cohort (*Heinzel S, et al. Annals of Neurology. 2021; 90, E1-E12*) reported a comprehensive analysis of the associations between microbiota composition and neurodegenerative markers. We have added this paper in the reference list and incorporated the interesting findings from general population in the discussion, especially the associations of microbial features with possible RBD (**paragraph 2, page 28**) and constipation (**paragraph 2, page 31**).

Comment 3: Little detail is given in the methods about patient selection- how was early PD defined, ie were participants within 3 years of PD diagnosis? The RBD-FDR group are defined as "high-risk" yet Figure 1 suggests they were sequentially recruited, and excluded if they had PD, cirrhosis, or colonic cancer. The authors should stipulate that no further stratification or exclusion was performed on the RBD-FDR group. Also, the diagnosis of probable and definite RBD should be defined in the methods and not just in Figure 1, unless I have missed this. What was the split between probable and definite RBD cases, and the rationale to lump them together in the analysis? Was a microbiome gradient seen from possible to probable RBD in the data?

Response 3: We appreciate the insightful comments. The description of subject selection, especially probable and definite RBD, may look unclear for readers who may be unfamiliar with RBD classifications. To address this issue, we have clarified the methods and summarized the selection criteria in **Supplementary Fig.1**.

In details, except for the common exclusion conditions in microbiota study (e.g., antibiotics use and severe gastrointestinal diseases), we incorporated additional subject selection criteria to simulate the gut-brain staging model of α -synucleinopathy. Definite RBD corresponds to the video-polysomnography (v-PSG) diagnosed RBD cases, while probable RBD refers to clinical interview diagnosed RBD.

As shown in the supplementary Fig.1, all RBD patients were v-PSG diagnosed and free of current neurodegeneration diagnosis. Early PD was defined as 1) the onset of parkinsonism less than 5 years, 2) with no dementia (i.e., a total score of Hong Kong version of Montreal Cognitive Assessment [HK-MoCA] > 21 and Clinical Dementia Rating [CDR] < 1); 3) had v-PSG diagnosed RBD, and 4) parkinsonism preceded by RBD symptoms.

With regard to RBD-FDR (first-degree relatives), we specifically excluded those FDR who were v-PSG confirmed RBD (Liu Y, Wing YK, et al. *Ann Neurol.* Apr 2019;85(4):582-592) and no further stratification was performed in this group of FDR subjects. While in the selection of control, those subjects with probable RBD, v-PSG diagnosed RBD or a positive family history of RBD symptoms were excluded. Overall, the severity of RBD symptoms progressively increased from control, RBD-FDR to patients with RBD and early PD, which could be reflected in the increasing scores of RBD questionnaire (**Supplementary Table 1**).

For the microbiome gradient from possible to probable RBD, we did not involve the concept of 'possible RBD' in this study, as possible RBD usually indicated RBD diagnosed based on the cut-off value of questionnaire and is used interchangeably with probable RBD by some authors (Mahlknecht P, et al. *Mov Disord.* 2015 Sep;30(10):1417-21; Zhang H, et al. *Neurology.* 2020;95(16):e2214). As for a potential microbiome gradient from probable to definite RBD, we only have a limited number of probable RBD cases (n = 11, RBD-FDR) in current study, which is apparently underpowered for further bioinformatic analysis. Nonetheless, we have performed a preliminary analysis in this reply letter. The data showed the abundance of two differential genera [*Eubacterium*]*_ventriosum_group* and *Collinsella* was significantly decreased (clr-abundance, -0.36 ± 1.8 vs 0.70 ± 1.6 , $p = 0.02$) and marginally increased ($0.67 -0.76 \pm 1.3$ vs -0.76 ± 1.3 , $p = 0.05$) in RBD-FDR with probable RBD (n = 11), respectively, when compared with control (n = 108), while no significant differences were observed between probable RBD and definite RBD patients (all $p > 0.05$). This finding further supports that PD-like gut dysbiosis occurs before the onset of RBD, at an even earlier prodromal stage. However, future studies with larger sample size are warranted to confirm the microbiota changes from probable to definite RBD cases.

Comment 4: How generalizeable are the author's findings to brain-first PD, that is in PD patients whose PD was not preceded by clinical or PSG-defined RBD? This is a potential limitation of the study, in that only half of the 36 early PD patients are in this category, and this should be discussed in the Discussions limitation section. This will be important in directing future therapies for PD patients.

Response 4: We agreed that the generalization of our findings to other subtypes of PD (brain-first vs body-first) was a potential limitation of the current study. In fact, we have clarified that all early PD patients were preceded by v-PSG diagnosed RBD (body-first subtype, **page 33-34**), and added this interesting debate issue (brain vs body first) in the Discussion section of the revised manuscript (**page 31-32**).

On the other hand, we believed that it is also a strength of our study that we proposed a staging model by taking into account of the heterogeneous conditions of PD (we only included ‘body-first’ subtype of PD) and prodromal PD. In other words, some of the inconsistent findings in previous microbiota studies (e.g., family Prevotellaceae, *Romano S, et al. npj Parkinson's Disease. 2021;7(1):27*) might probably be due to the absence of further stratification of PD patients. Nevertheless, future studies are needed to confirm the associations of gut microbiota with inclusion of brain-first PD and other types of α -synucleinopathies (e.g., dementia with Lewy bodies), which could further facilitate the implementation of biological subtyping of α -synucleinopathies and is critical for clinical trials involving neuroprotective therapies (*Berg D, et al. Nature Reviews Neurology. 2021;17(6):349-361*).

Comment 5: Did the authors look at parallel measures of G.I mucosal inflammation or integrity from the faecal samples? This would be very interesting, to directly link changes in the microbiome to mucosal breakdown

Response 5: We fully agreed with the reviewer that linking microbial changes to mucosal inflammation and gut permeability would provide insights into the direct impact of gut dysbiosis on the GI tract, and even the pathogenesis of RBD and Parkinson’s disease. Although we have not yet done this important work at this stage, it is part of our future plans for this important staging model. We have addressed the necessity of including inflammatory markers in the discussion section of this manuscript (**paragraph 2, page 32**).

Comment 6: The manuscript is well written, but would benefit from a general English review as I identified a number of minor language and figure errors:

Abstract: suggest change to:"There has been ongoing debate about the causal relationship between gut dysbiosis and Parkinson’s disease (PD), partially due to the high variability of gut microbiota (sensitive to host lifestyle and physiological characteristics), and the difficulties in identifying subjects with a prodromal PD syndrome"

Response 6: Thanks for the suggestion. We have revised the manuscript in more details on the language and grammatical aspects including the abstract.

Comment 7: Introduction para 2: change language and insert definition of earlier or prodromal RBD:"Patients with isolated/idiopathic RBD reported an increased

prevalence of constipation, and show increased phosphorylated α -syn immunostaining in ENS. Likewise, PD patients with premotor RBD features appeared to exhibit prominent degeneration of peripheral nervous system (e.g., increased constipation and enteric α -syn histopathology) when compared to those without, suggesting that PD with RBD is a distinct subtype of Parkinson's disease reflecting the gut-brain hypothesis of α -synucleinopathy. On the other hand, isolated RBD features, e.g., isolated dream-enactment behaviors not meeting RBD polysomnographic diagnostic criteria, might reflect a prodromal stage of RBD and the early presentation of α -synucleinopathy..."

Response 7: Thanks for the suggestion. We have made clearer descriptions of the concept of staging model in the paragraph 2 of Introduction section (**page 4**). For the definition of prodromal RBD, we explained it as "isolated RBD symptoms, but not yet meeting the v-PSG diagnostic criteria for RBD, might reflect a prodromal stage of RBD and the early presentation of α -synucleinopathy", as according to the review paper of RBD (Hogl B, et al. *Nat Rev Neurol.* 2018;14(1):40-55) and latest diagnostic guidelines from the International RBD Study Group (Cesari M, et al. *Sleep.* 2022;45(3)).

Comment 8: Intro: para 3, last sentence- suggest reference here TREND study which showed that individuals with possible RBD identified on screening questionnaire showed altered beta-diversity in microbial composition. DOI: 10.1002/ana.26128

Response 8: We appreciate the reviewer's suggestion. The TREND study has been referenced in the first sentence of revised Introduction paragraph 3 (**paragraph 2, page 5**): "To the best of our knowledge, only three prior studies reported gut microbiota in v-PSG diagnosed RBD (n = 21 and 26, respectively) [22, 23] and possible RBD as assessed by screen questionnaire (n = 84).[24] suggesting similar trends of changes in RBD and PD patients".

Comment 9: Figure 4-Differential gut microbes at the early stages of α -synucleinopathy-The use of 2 similar blue colors to denote RBD FDR and RBD means they are very difficult to distinguish in Figure 4. Suggest change

Response 9: We thank the reviewer for his/her suggestions on Figure 4 (changed to Fig.3 in the revised version of manuscript). We hope that the reviewer will agree that the current gradient of the blue colour has been adjusted to make the differences between groups more discriminative.

Comment 10: Discussion para 3, change to:"Interestingly, a previous study showed that this disruption effect of *Akkermansia* and other mucin-degraders could be reversed by switching to a high-fiber diet, indicating the potential of diet therapy in intervening gut dysbiosis at early α -synucleinopathy" to minimise confusion

Response 10: Thanks for the reviewer's suggestion. In the revised version of manuscript, we found that the abundance of *Akkermansia* was significantly increased in patients with RBD and early PD as compared with control when adjusting age, sex, and family cluster. In addition, host-microbiome interaction analysis showed that the enrichment of *Akkermansia* in RBD and early PD was also highly related to age, constipation symptom, and antidepressant use. Based on this result, we further revised the discussion according to the reviewer's suggestion as follows (**paragraph 1, page 30**): "Interestingly, there is a possibility that the disruption effect of *Akkermansia* and other mucin-degraders might be reversed by switching to a high-fiber diet, indicating the potential of diet therapy in intervening gut dysbiosis at early α -synucleinopathy, especially among patients with concomitant anxiety and depressive disorders.[46]."

Comment 11: Discussion- last para, does not read well. Suggest consider changing to below or similar:"In conclusion, our study suggested that gut microbiota might be a promising target for the early prevention of PD in those with polysomnographically diagnosed RBD and their first degree relatives who manifest earlier prodromal forms of RBD. Future prospective studies, combined with investigations of gut metabolism, inflammatory markers and enteric α -syn pathology are required for a thorough understanding of the role of gut microbiota in the pathogenesis of α -synucleinopathy. "

Response 11: We appreciate the reviewer's suggestions and have revised the conclusion paragraph accordingly.

Comment 12: References: Ref 33 appears incorrect or incomplete

Response 12: We have checked the citation format of reference 33 in PubMed and corrected it in the revised version of manuscript (as reference 57).

Reviewer #2 (Remarks to the Author):

Comment 1: The article from Dr Bey Huang et al describes the results of a study aimed at characterizing the gut microbiota in the early stage of a-synucleinopathies. They performed a cross-sectional study in which they collected and analysed stool samples of controls, first-degree relatives of RBD patients, RBD patients, and early PD patients. They show that the alterations in gut microbiota composition observed in PD, especially the depletion of SCFA producers, appear already in RBD patients, suggesting a causal link between gut microbiota and PD.

First, I would like to congratulate the author on this really nice and novel piece of work. The study in my opinion is well planned, the overall recruitment and sampling strategies are well described and depicted. I think that overall, the analyses are well performed, and I particularly appreciate the creativity of the authors in presenting the data.

I think the paper deserves publication, but I have some concerns that I would appreciate be addressed before final acceptance.

Response 1: We thank the reviewer for summarizing the key findings and acknowledging merits of this study. We responded to the issues and concerns raised by the reviewer as follows.

Comment 2: General comments: It is not clear to me how the authors analysed the 16S data. Qiime2 is a sort of wrapper program that uses other programs to analyse the data. Were OTUs or ASV called? How was this done? Which program was used? For example, “Sequencing tags and contaminant reads were removed by alignment tools, samples with low total frequency”, what alignment tools? Name them, with version and reference.

Response 2: According to the suggestions from the reviewers, we made a clearer description of the 16S data analysis in the Online Methods (**page 35-36**). In this revision, we explained how to denoise paired-end sequence data into ASVs table (via DADA2), filter out low-quality data (via q2-feature-table), and predict the taxonomic affiliation of each ASV (via q2-feature-classifier). The name, version, and reference (if necessary) of each QIIME 2 plugin were indicated in the revised methods accordingly. We hope that these details may help to clarify the analysis of 16S data.

Comment 3: The statistical description of the approaches used for analysing the microbiome can be greatly improved. Please, provide the details of the model structures, report versions of the packages, and include references to them. For example, explain why you choose to use GEE instead of LM. Also provide detail of how the GEE models were run, what type of data/correlation structure you used within the model.

Response 3: We thank the reviewer's comment on statistical description. We have made major revisions on more detailed description of statistical methods (**page 36-40**). The model structures, versions of packages, and references were provided more clearly accordingly. The use of GEE model was adapted from our case-control-family study (*Liu Y, et al. Ann Neurol. 2019;85(4):582-592*), at which we considered the potential correlations (e.g., genes and family environment) between FDRs within the same family. The GEE model has been widely used to analyse family-clustered data (*Ziegler A, et al. Biometrical Journal. 1998;40(2):115-139*). In particular, one previous study reported that "GEE models are a flexible regression-based approach for dealing with related data that arises from correlated data such as family data" and had "more robust findings and produced more reliable parameter ..." (*Homish GG, et al. Addict Behav. 2010;35(6):558-63*).

In addition, in responding to the reviewer's suggestion, we have made clearer descriptions about GEE model in the section of Online Methods (**paragraph 2, page 36**): "Considering the potential association between subjects from the same family, we applied Generalized Estimation Equation (GEE) model to adjust family clustering in the comparisons of gastrointestinal features, clinical characteristics, and neurodegenerative markers among groups.[21] GEE model was constructed with *family id* as the cluster factor and *subject id* as the within-cluster factor, using an independent correlation structure. The type of model was specified as Linear (continuous variables) or Binary logistic (binary variables) based on the distribution of the dependent variables."

Comment 4: The authors declare in the reporting summary that no custom code was created. I am afraid I disagree. The authors had to manipulate their data, filter them, fit models based on their experimental design, extract estimates, errors, and p-value all tailored to their dataset. In my opinion, this is custom code. I believe the authors need to deposit their code in a public repository to foster transparency and reproducibility. Similarly, the authors have to provide a link to the raw reads deposited in NCBI/ENA. I believe this should be done at this stage, as the data can be made private and reviewer access can be requested.

Response 4: We appreciate the reviewer's suggestion. Code for the bioinformatic analysis in R program was deposited in GitHub (<https://github.com/Joannehb/Gut-microbiota-across-early-synucleinopathy>). GEE models and basic statistical analysis in SPSS was described in the Online Methods to ensure the reproducibility of the data. Besides, we deposited the raw sequencing data in the European Nucleotide Archive (PRJEB52086), which has been made public in July 2022.

Comment 5: A less important point, more strength to the findings could be given by citing additional recent meta-analyses that reported depletion of SCFA producers in PD and enrichment of Akkermansia and Christensenellaceae (e.g. Shen et al 2021, Romano et al 2021, Toh et al 2022).

Response 5: We thank the reviewer's suggestion. We added the meta-analysis papers mentioned by the reviewer (*Romano S, et al. npj Parkinson's Disease. 2021;7(1):27; Shen T, et al. Front Aging Neurosci. 2021;13:636545; Toh TS, et al. Parkinsonism & Related Disorders. 2022/01/01/ 2022;94:1-9*) in the revised version of manuscript.

Comment 6: The DMM model used to identify the community types is not convincing for me. Similar approaches have been used to identify enterotypes and have been highly debated. In my opinion and experience, these approaches will provide a set of more or less robust clusters even though they have no biological meaning or are not significantly different. For example, in Figure 3A from the PCoA plot, I can see that in terms of dissimilarity distances the Type B and C are basically identical. D and E might be slightly different but also largely overlapping. Can please the authors provide additional evidence that the clustering is robust and meaningful.

Response 6: We have carefully considered the reviewer's suggestions with regard to the community component analysis. As noted by the reviewer, five community types identified by DMM approach were not clearly separated from each other in principal coordinates analysis (PCoA) plot, albeit pairwise comparisons in the PERMANOVA analysis indicated significant differences. We also fully agreed that the biological meaning of these community types was largely unknown, and could not be further explored in depth in the present study. Besides, we have studied the clustering pattern of gut microbiota at different early stages of α -synucleinopathy using PERMANOVA test and PCoA analysis. The results showed a significant microbial shift in patients with RBD and early PD when compared with control and RBD-FDR (all q-values < 0.001) (**page 10-11**). In the light of the above, we decided not to include community component analysis in the revised version of manuscript.

Comment 7: Similarly, they report in the Discussion that “The community type D resembled an unstable community type in Human Microbiome Project (HMP) 23 (supplementary table 10).” This seems to be based on checking what bacterial taxa (phyla) are mainly present in the description of these groups. Can the author provide any evidence that these are actually similar communities? I am afraid that a comparison just based on the cluster broad description is not enough. For example, in type D Faecalibacterium is low in abundance. However, it is also a Firmicutes, which is a group of bacteria enriched in the Type B community of Ding et al.

Response 7: We agreed with the reviewer that the similarity of community types should not be determined solely on the basis of selected taxa abundance. As elaborated above in the response to reviewer’s comment 6, we have deleted the community component analysis and relevant discussion from this revision.

Comment 8: The authors mention a “pairwise comparison for the global composition of the gut microbiome” in pg 9 and afterward. How was this done? The permanova analysis, presumably done using the adonis2 function in vegan (as this was not specified) is not a pairwise test. There is no mention of this in the method section.

Response 8: We agreed with the reviewer that pairwise analysis is not an accurate description. In the revised manuscript, the PERMANOVA analysis was described as follows (**paragraph 2, page 37**): “Compositional differences between each pair of groups were analysed using permutational multivariate analysis of variance (PERMANOVA) in the vegan R package (“adonis2” function), [68] and distance matrix for “adonis2” function was constructed based on the Bray-Curtis distance of the relative abundance of whole genera (“vegdist” function)”.

Comment 9: Also, in my opinion, Fig 2 is not well presented. I would first present an overall PCoA with all samples and report the pairwise comparison in a table. I would remove figure 2B. Fig 2A is very nice, and I will present it as Fig 2B.

Response 9: We appreciate the reviewer’s suggestions. In this revision, fig.2 (**page 11-12**) is made up of (A) a PCoA plot of whole samples with group centroids indicated, (B) only the group centroids with error bars are shown in PCoA plot, and a separate table summarizing the pairwise comparison results. We hope that the reviewer will agree to our current figure revision.

Comment 10: Just a minor note here, the authors mention that they tested for homogeneity of variance in the groups, however from the PCoA seems the Early PD and RBD_hr have much broader dispersion of points. Might this also explain why there is only a marginal difference between RBD_hr and RBD_lr? Also, where is the comparison between RBD_hr and PD?

Response 10: Thanks for the astute observation. The microbial variation is greater in RBD (include low- and high-risk) and early PD than in control and RBD-FDR. Heterogeneity in group dispersion may affect the statistical analysis of overall microbial compositions (*Anderson MJ, et al. Ecological Monographs. 2013;83(4):557-574*). Therefore, we added the discussion of dispersion in the revised section of Results (**paragraph 2, page 10**):

“Homogeneity of dispersion test indicated that RBD and early PD had higher levels of variation (all $q < 0.05$, Supplementary Fig.2A), which may influence the PERMANOVA results, especially in early PD group (smaller number of subjects).[28] Nevertheless, microbiota shift at early α -synucleinopathy would still be strongly supported by principal coordinates analysis, at which the first two principal components accounted for 11.78% of total variation (Supplementary Fig.2B). Using inter-group comparisons along first (x axis) and second principal components (y axis), we confirmed the similar and different clustering patterns of early PD versus RBD and control, respectively (Fig. 2A).”

In this revision, microbiota changes between control and RBD subgroups were moved to the supplementary materials. Patients with RBD were divided into different subgroups based on 1) the calculated probability of prodromal PD and 2) disease duration of RBD. Still, there are significant greater dispersion in RBD subgroups as compared with control (ANOVA post-hoc, q -values < 0.05). The detailed results are shown in **Supplementary Fig. 5C-D**.

Comment 11: I believe that the authors should do a little more effort in explaining how they performed the differential abundance analyses (DA). ANCOM accounts for the compositionality of the data. However, GEE will not. Were the data rarefied and transformed? How was this done?

Response 11: We thank the reviewer’s suggestion. We have made major revisions to the differential abundance (DA) analysis in the revised section of Online Methods (**page 37-38**). The details of input data for each analysis, including data filtering (e.g., prevalence and abundance filtering) and transformation approach (e.g., relative abundance and centered log-ratio trans), were reported accordingly.

Comment 12: Also, why did they choose a cutoff for the ANCOM of 0.6? As a minor comment, ANCOM might have a high false-negative rate, especially in the first version the author used. So, even if not critical, the authors might consider using an updated versions or additional methods.

Response 12: We chose 0.6 as the cut-off value for ANCOM because a lower cut-off was recommended when one would like to “[explore more discoveries \(larger power\)](https://github.com/FrederickHuangLin/ANCOM-Code-Archive)” (<https://github.com/FrederickHuangLin/ANCOM-Code-Archive>). In this study, we primarily used ANCOM to screen potential differential taxa, while the significance of inter-group differences in taxa abundance was further examined in GEE model.

Nonetheless, we agreed with the reviewer’s suggestions to use an updated ANCOM or additional method for the differential abundance analysis. In the revised manuscript, we have changed the DA analysis methods to Kruskal-Wallis and MaAsLin 2, to avoid an overly high false discovery rate or false negative rate respectively. (*Nearing JT, et al. Nature Communications. 2022;13(1):342*). The details of DA analysis were elaborated in the section of Online Methods (**page 37-38**).

Comment 13: The data presented in Fig 5 and 6 are the ones that concern me the most. As far as I could understand the ML was built using the differential abundant taxa identified in the previous ANCOM analyses. However, these taxa have been selected on the whole dataset. This means that they will be selected on both the training and test dataset of the ML model. In my opinion this will generate some “data leakage” and inflate the AUCs. Can please the author clarify why this should be acceptable and reperform the RF model (if needed) on the whole dataset or at least perform variable selection only on the training set?

Response 13: Thanks for the comments and suggestions about machine learning. We agreed that current selected features in the ML model was theoretically “trained” from the whole dataset, although it was not directly determined by the ML feature selection method. Our initial interest in using this approach was to confirm that the differential taxa were also powerful in distinguishing disease status from the control group. Nevertheless, this is not the standard method for constructing ML prediction models. To address this issue and to accommodate the suggestion of hold-out method from another reviewer, we performed variable selection (recursive feature elimination via “caret” R package, 25 repeats of 10-fold cross-validation) from all genera (abundance > 0.05% and prevalence > 10%, n = 88) on the training set (70% of whole dataset). Then the performance of the fit model was examined on

the independent test set (30% of whole dataset). The relevant descriptions have been revised and added to the section of Online Methods (**page 38-39**) and Results (**page 17**).

Comment 14: I do not fully understand fig 6A. Was this done using a permanova (as I can read in pg 29)? Where are the p-values? Permanova on the whole dataset will not report R2 for each group. Please clarify.

Response 14: We thank the reviewer's suggestion. . In the revised version of manuscript, PERMANOVA analysis of covariates were specified in the section of Results (**paragraph 1, page 19**): “Multivariate statistic models (PERMANOVA, 99,999 permutations) were built including each pair of groups and all covariates, while PD specific drugs (Carbidopa/Levodopa, MAO-B inhibitors, and dopamine agonist) were only assessed in the comparisons involving early PD group.” and the data was described in **Fig.5B (page 21-22)**.

Comment 15: Similarly, they write that they stratified the data by various confounding factors and then re-assessed whether SCFA taxa remained DA across groups, The conclusion was that “In general, the observed microbial changes as calculated by PERMANOVA test, GEE model and ROC analyses remained robust in RBD patients after further stratification of confounders”. However, it seems to me that this is not always the case and that sometimes an interaction is actually present in the data. What happens if the authors test whether the interaction between RBD:host-factors is actually defining taxa abundances? How should this be interpreted? I have the feeling that the real picture is more complex than what is presented. I would have liked to see also that the authors accounted for the confounding effect of medication on the SCF taxa abundance. It seems to me that RBD and PD took more antidepressants than the rest of the participants. Is this drug confounding taxa abundances?

Response 15: The reviewer raised the concerns that the true host-microbiome interactions, especially the impact of host factors on taxa abundance, and the influences of diverse clinical drugs, were under investigated. We fully agreed with the reviewer's comment and have revised our presentation of host-microbiome interactions by using PERMANOVA (overall composition) and MaAsLin 2 analyses (individual taxon abundance). Accordingly, we made a clearer description of the potential impacts of various host factors on gut microbiota in the section of Results (**page 19-20**): “In addition to the grouping factor, other covariates including BMF score, sex, osmotic laxatives, and proton pump inhibitors (PPIs) uses exhibited strong associations with microbial composition, while age, statin, antidepressant, benzodiazepines and PD specific drugs had minimal influences on compositional changes

(Fig. 5B, Supplementary Table 4)..... the associations between BMF score and taxa abundance were extensive and resembled that of disease progression. In particular, a higher score of BMF (i.e., slower colon transit) was strongly correlated with a lower and higher abundance of *Butyricoccus* ($\beta = -0.34$, q-value = 0.009) and *Oscillospiraceae* UCG-005 ($\beta = 0.32$, q-value = 0.003), respectively (Fig. 4C). With regard to the drug effect, the use of antidepressant, statin, and osmotic laxatives seemed to have greater impacts on differential taxa than PPIs, benzodiazepines, and PD specific drugs. In particular, we observed a strong co-occurrence of antidepressant usage and RBD/early PD-enriched genera (e.g., *Akkermansia* and *UBA1819*) in our data.”

Comment 16: Also, it seems to me that in Fig 6 only data for RBD are shown. However, they mentioned that they did these analyses for RBD-FDR as well. Where are these data? The supplementary table would be clearer if they showed the results before and after stratification side by side.

Response 16: We thank the reviewer’s suggestion. The original Fig. 6 has been removed from this study due to the change in methodology. In the revised supplementary tables, we made a clearer classification of different bioinformatic analysis. Besides, detailed results for each analysis that were not reported in the main text were provided in the relevant supplementary tables/figures.

Comment 17: Finally, how “hard stool” was defined? Using the Bristol chart? Was this included as a binary variable in the model? Wouldn’t be more appropriate to use the Bristol scores, or even better the water content of the stool?

Response 17: The presence of “hard stool” or “constipated stool” was defined as “type 1: separate hard lumps, like nuts (hard to pass)” or “type 2: sausage-shaped but lumpy” in the Bristol Stool Form Scale (BSFS), which was consistent with the classification in Rome IV Diagnostic Questionnaire (R4DQ) (*Whitehead WE, et al. ROME Foundation; 2016*). The original BSFS score was used as a continuous variable in the comparison of constipation symptoms between different groups (**Supplementary table 1**). However, as a potential confounder assessed in sensitivity analysis, we considered it more appropriate to use a binary variable, e.g., the presence or absence of “hard stool”.

For the stool water content, we did not directly measure the water content in this study, because it was not the interest of this study in the designing phase. While BSFS score could be a good proxy of it (*Vandeputte D, et al. Gut. 2016;65(1):57-62*), we agreed with the

reviewer that an objective measurement of stool water content is much better and could be involved in a future study.

Comment 18: I liked the mediation analysis very much. I think it would be very helpful if the authors could explain this procedure better to make it more accessible to the non-experts.

Response 18: We thank the reviewer for affirming and supporting the mediation analysis. The description of the basic concept and key variables in interpreting the results have been added in the section of Online Methods (**paragraph 3, page 39**): “The average causal mediation effects (ACME) indicated the indirect effect that goes through the mediator, while average direct effects (ADE) described the direct effect of exposure on the outcome. We tested the significance of this indirect/direct effect using bootstrapping (10,000 resamples) procedures. The proportion of the effect of the exposure on the outcome that goes through the mediator was evaluated by dividing the ACME by the total effect (ACME + ADE)”. Besides, the original code for the analysis was deposited together with other bioinformatic analysis in the GitHub (<https://github.com/Joannehb/Gut-microbiota-across-early-synucleinopathy>).

Conclusions:

The paper is of high quality.

I believe it cannot be published in this form.

Thorough adjustment of the method description, verification of the DMM, ML, and confounding analyses, and deposition of raw reads and R code in public repositories are needed.

Reviewer #3 (Remarks to the Author):

Comment 1: This is a timely and novel study of the potential changes in the fecal microbiome of subjects who are otherwise healthy controls (n=108 in final data set), or have a diagnosis of REM sleep behavior disorder (RBD; n=170), are first degree relatives of a subject with RBD (RBD-FDR; n=127), or have early-stage Parkinson's disease with a pre-motor symptom phase history of RBD-like symptoms (n=36). The study uses standardized clinical measures and instruments to ascertain and assess the subjects, and standard methods for performing 16S-based metagenomic sequencing analysis. They report a number of interesting findings, such as overall shifts in the microbiome composition in subjects with RBD who are at-risk for PD, and subjects with

early stage PD compared to controls and RBD-FDR groups. Moreover, they also were able to identify 5 phylotypes based on shifts in composition determined by Dirichlet multinomial mixture modeling and observed that the subject groups differed in the phylotype they were most strongly associated with. For example, type D (associated with lower *Bacteroides*, *Faecalibacterium*, and higher *Bifidobacterium*) was more common and types A and B less common in RBD and early stage PD compared to controls and RBD-FDR. Further analysis by the authors revealed two specific short-chain fatty acid (SCFA) producing taxa - *Rosburia* and *Lachnospiraceae* - were reduced in all groups compared to controls, while one mucin-degrading bacteria was present at higher abundance in all groups compared to controls. The authors went on to assess the extent to which a diagnostic classifier could be developed based on the data for these and other studies, and claim to have achieved an ROC of 0.80 for distinguishing RBD subjects from controls, when using a classifier with 15 genera and the likelihood ratio for PD (exclusive of RBD-specific questions). Finally, the authors report some interesting associations between host activities and microbiome shifts after controlling for a number of potential confounds that they identified.

Response 1: We thank the reviewer for his/her careful review and insightful comments. We responded to the issues and concerns raised by the reviewer as follows.

Comment 2: Overall, the study provides some convincing findings that would be of interest to many in the movement disorder field. However, there are a number of significant concerns that reduce overall enthusiasm. First, as a clinical study, although the ascertainment methods appear sound, the composition of the subject groups is markedly different in many ways that could impact study outcomes considerably in a non-specific manner. For example, the RBD-FDR group is approximately 10 years younger than all other groups and notably more female.

Response 2: The reviewer raised the point that statistical correction might not overcome all the effect of group differences on the study outcomes. We fully agree with the reviewer that host variables (e.g., age and sex) could confound gut microbiota studies of human disease in unpredictable manners (*Falony, G., et al. Science. 2016; 352(6285): 560; Vujkovic-Cvijin I, et al. Nature. 2020;587:448-454*). The reasons for us to establish the staging model with varying group compositions, and how we will clarify the specific impact of host factors, are further described as follows:

First, this is a cross-sectional study that encompass different early stages of α -synucleinopathy (i.e., a quasi-longitudinal design). Thus, as a progressive disease with long

prodromal period (as long as 20 years) (Bloem BR, et al. *The Lancet*. 2021;397(10291):2284-2303), there would be some differences in the compositions of groups, especially in age and healthy status. As for the unmatched sex between four groups (more female subjects in RBD-FDR), we have recognized this as a limitation of the study and specified in the Discussion section (**paragraph 1, page 32**): “Fourthly, RBD-FDR, which reflected an even earlier prodromal stage, were composed of younger and more female subjects. Nevertheless, the key findings remained robust even after adjusting for these sociodemographic differences.” On the other hand, the inclusion of a younger RBD-FDR group, who shared genetic risk factors with RBD and presented increased RBD/constipation features (Liu Y, et al. *Ann Neurol*. 2019;85(4):582-592), is an important and innovative part of this study. As noted in the Introduction (**paragraph 2, page 5**), a study design that incorporate different prodromal phases will help to delineate gut microbial alterations as the disease progresses, and further contribute to the understanding of disease pathophysiology and potential neuroprotective therapies. Interestingly, our data showed that RBD-FDR has developed PD-like gut microbial alterations (e.g., enrichment of *Collinsella* and depletion of *[Eubacterium]_ventriosum_group*) when compared with control, despite being nearly 10 years younger than both controls and PD patients.

To clarify the specific impact of host factors on study outcomes, we have further revised the statistical methods. Multivariate association analyses (PERMANOVA, MaAsLin 2) were used to study host-microbiome interactions at the level of overall composition and individual taxa/metabolic pathways (**page 19-22, page 25-27**). Current data showed that compositional shift of microbiota at early α -synucleinopathy could be affected by sex differences between the groups, but far less than the effect of constipation symptoms (e.g., bowel movement frequency) (**Fig. 5B**). The influence of age on differential taxa is minor, but male is related to a higher abundance of genus *Collinsella* (**Fig. 5A**). Since RBD-FDR consist of more females than control, increased *Collinsella* in RBD-FDR could hardly be explained by the sex difference between the groups. To the best of our knowledge, there is no previous study that reported age and sex-dependent changes of *Collinsella*. Only one study found that *[Eubacterium]_ventriosum_group* negatively associated with age (Wei Z-Y, et al. *bioRxiv*. 2020:2020.03.30.015305), which seemingly differed from the findings in this study.

Furthermore, in this reply letter, we performed additional association analyses within the control and RBD-FDR groups. It was found that age was not related to the abundance of *Collinsella* and *[Eubacterium]_ventriosum_group* in healthy controls (**Figure 1 of this reply letter**). Consistently, we observed that male control subjects had higher level of *[Eubacterium]_ventriosum_group*, while this change cannot be captured in RBD-FDR group (**Figure 2 of this reply letter**). Combining the findings from multivariate models, we

concluded that there was a potential association between *[Eubacterium]_ventriosum_group* and sex, while this interaction did not significantly confound the observed microbial changes in this study.

Figure 1 (in this reply letter). Spearman correlation analysis showed no significant relationship between age and centered log-ratio-transformed abundance of differential taxa in control (n = 108).

Figure 2 (in this reply letter). Comparisons of centered log-ratio-transformed abundance of differential taxa between male and female subjects in control (n = 108, A-B) and RBD-FDR (n = 127, C) groups by using Mann-Whitney U test.

Comment 3: The RBD group also has twice as high a rate of PPI (proton pump inhibitor) and antidepressant use as well as statin use than the other groups, and the PD group has much higher use of dopamine-acting drugs. All of these things are not themselves part of the core pathogenetic mechanisms in RBD or PD, but they all are known to impact microbiome composition and gut function. Thus, it is hard to accept

that some of the differences observed are not due to such non-specific influences on their data.

Response 3: We thank the reviewer for pointing out the importance of investigating confounders in microbiota studies. This was also our primary concern in the statistical analyses, and therefore, we have conducted rigorous analyses to examine the host-microbiome interactions. For example, the selection of potential confounders was based on significant group differences (a common practice in studies where the number of cases does not significantly exceed the number of variables), and the effect of true confounders were further tested in the sensitivity analysis. However, as noted by the reviewer, we might have overlooked the impact of some strong confounders (e.g., commonly used drugs) that were only marginally different between the groups (*Vich Vila A, et al. Nature Communications. 2020;11(1):362*). Thus, in the revised manuscript, we have added more potential confounders, including PPIs, stains, and PD-specific drugs, in the multivariate analysis. Current results showed that bowel movement frequency (BMF) score, sex, PPIs, and osmotic laxatives could influence overall microbial composition, while BMF score, age, sex, antidepressant, osmotic laxatives, statin, PPIs, and benzodiazepines may impact taxa and pathway abundances. Nevertheless, gut microbiota changes, including the increase of pro-inflammatory *Collinsella* and depletion of butyrate-producing [*Eubacterium*]_ventriosum_group, remained robust at early stages of α -synucleinopathy even after adjusting all the potential confounders. The detailed descriptions are shown in the Results section (**page 19-20, page 23, Fig.5, Fig.7, Supplementary table 10-12, 15**)

Comment 4: Second, as a microbiome study, the methods and depth of analysis is somewhat superficial. It would have been very helpful for the authors to perform more sophisticated metabolic pathway profiling based on the compositional shifts seen between their groups, rather than make somewhat sweeping statements based on a select few number of taxa.

Response 4: We thank for the reviewer's suggestions. Function prediction from 16S marker sequences have been widely used in microbiota study and could be an exploratory and preliminary analysis of gut metabolic profile when metagenomic sequencing is not feasible. Accordingly, the results of metabolic pathway abundance changes have been added to the revised version of manuscript (**page 25**). The new data indicated that microbial fatty acids metabolism (short chain fatty acids to lactate and ethanol) was increased, while preQ₀ biosynthesis (7-deazapurine biosynthesis) was decreased in RBD-FDR, RBD and early PD after adjusting all potential covariates. In addition, the salvage and de novo pathways of

vitamin B₁₂ biosynthesis were significantly enriched in control than in RBD-FDR group (Fig.7).

Comment 5: Third, as a machine-learning diagnostic classification study, the work is simply not valid enough to draw any strong conclusions from. When evaluating the performance of a machine-learning tool, the use of 4-fold cross validation should not be viewed as an independent test of the performance of the model, but rather just a validation that an optimized model was developed. A true hold out set of fully-independent samples that were never used to help train the model must be used to establish its performance. There are clear standards of practice for the adoption of such approaches in precision medicine.

Response 5: The reviewer raised the concerns that no completely independent test set was included in ML prediction models in this study, and therefore could not validate the generalization performance of microbial classifiers in predicting early stages of α -synucleinopathy. We fully agreed that the lack of separate test dataset had a substantial impact on the certainty of results. In the revised manuscript, we re-performed the whole machine-learning analysis in compliance with standard practice. The original dataset was divided into training (70%) and test (30%) sets, and the performance of optimal model was validated in the test dataset. Please kindly refer to the detail responses in the previous section to reviewer 2 and the revised random forest method in this manuscript (page 38-39).

Comment 6: Finally, from a statistical power point of view, the study simply has too few subjects to conclude that the results would be robust and reproducible in other cohorts. While RBD subjects and their relatives are somewhat difficult to find and recruit into studies, there are ample subjects of subjects with early stage PD.

Response 6: We agreed with the reviewer that a larger number of subjects would always pose an additional advantage to the statistical power and increase the reproducibility of study outcomes. The reasons for us to include relatively small size of PD patients (n = 36) in this study included the followings:

First, the number of subjects required for early PD was based on the “Sample size estimation” in the proposal of this study (Health and Medical Research Fund of the Food and Health Bureau of Hong Kong, China [05162876], 2016-17) at which the current study was partially supported. In the proposal, we have estimated that “According to a previous study comparing the gut microbiota composition between patients with PD and controls (Scheperjans F, et al.

Mov Disord. 2015;30(3):350-8), the effect sizes for various bacterial families are quite large, with a Cohen's d greater than 2.68.” Based on this effect size, a sample size of 36 for PD group could achieve a power over 0.95 (type I error of 0.05) in Wilcoxon-Mann-Whitney test (two groups).

Second, as indicated in the Introduction (**paragraph 2, page 3-4**) and Online Methods sections (**paragraph 2 and 3, page 33-34**), there were some strict selection criteria in recruiting PD patients. All of them would need to fulfill: 1) a diagnosis of PD confirmed by neurologist according to the United Kingdom Parkinson's Disease Survey Brain Bank; 2) parkinsonism less than five years (average 3.7 ± 1.3 years); 3) no dementia; 4) had video-polysomnography confirmed RBD; and 5) parkinsonism preceded by RBD features as assessed by clinical interview. Therefore, this group of PD patients would represent a subtype of PD (“body-first”) reflecting the gut-brain hypothesis of α -synucleinopathy (*Berg D, et al. Nature Reviews Neurology. 2021;17(6):349-361*). As a result of all these restrictions and selection criteria, the recruited early PD subjects in this study may represent a more ‘homogenous’ subtype of PD (body-first).

Finally, the aim of this study was to confirm whether PD-like gut dysbiosis has occurred at earlier and prodromal stages of disease (i.e., RBD and RBD-FDR), rather than conduct a comprehensive investigation of PD-related microbial features, as the latter has been well characterized in previous studies (*Keshavarzian A, et al. Mov Disord. 2015;30(10):1351-60; Hill-Burns EM, et al. Mov Disord. 2017;32(5):739-749; Qian Y, et al. Brain. 2020;143(8):2474-2489*). Nevertheless, we agreed that a more balanced design of study (similar number of cases in each group) would benefit statistical analysis, and probably generate clearer trend of changes in microbiota with the progression of α -synucleinopathy (e.g., some genera were marginally increased/decreased in early PD as compared with control). In the revised version of manuscript, we have specified this as a limitation of study in page 31-32: “Secondly, the sample size of early PD was relatively modest. Nonetheless, we still observed significant and consistent changes of gut microbiota in PD as similarly reported in previous studies. Thirdly, this study focused on the gut-to-brain staging model of PD (“body-first”), therefore, additional research is needed before we can generalize the findings to the prodromal stages of other PD subtypes (e.g., “brain-first”)”.

Comment 7: Other minor concerns that merit correction include confirming whether the UPDRS was used instead of the UK Brain Bank criteria for verification of PD disease status in living subjects (page 25).

Response 7: The diagnosis of Parkinson's disease was confirmed by experienced neurologist according to the UK Brain Bank criteria (we have updated the reference to *Hughes AJ, et al. J Neurol Neurosurg Psychiatry. 1992;55(3):181-4*), which has been reported in our previous publications (*Zhou J, et al. Sleep. 2016;39(8):1543-50; Liu Y, et al. Sleep Med. 2019;56:128-134; Feng H, et al. 2020;88(4):817-829; Wang J, et al. Mov Disord. 2020;35(11):2077-2085*).

The total score of UPDRS part III was used to reflect the severity of motor dysfunctions at different early stages of α -synucleinopathy, which was further incorporated into the estimation of the total likelihood ratio of prodromal PD (*Liu Y, et al. Ann Neurol. 2019;85(4):582-592; Wang J, et al. Journal of Neurology, Neurosurgery & Psychiatry. 2021:jnnp-2021-327460*).

Comment 8: Whether any correction for multiple testing was performed on individual taxa data, or simply the nominal significance reported throughout.

Response 8: Yes, multiple comparisons were corrected using Benjamini-Hochberg method in this study, as specified in the Online Methods sections (**page 36, 38**). In the section of Results, we reported q-values (i.e., corrected p-values) in analyses involving multiple testing, such as the differential abundance analysis (**page 13**).

Comment 9: And inclusion of sequencing and quality assurance criteria parameters, such as the percentage of aligned reads, average coverage, length of trimmed reads, etc. One would like assurances that differences were not influenced by inherent biases in the raw sample material.

Response 9: We agreed with the reviewer that sequencing quality control was critical for gut microbiome study. In the revised version of manuscript, we have specified all the parameter settings in the 16S data processing at QIIME2 platform (**page 35-36**). Besides, the exclusion of low-quality data (read count < 1000) and the use of compositional data (e.g., centered log-ratio transformation) in this study could further help reduce the biases in the raw sample materials.

REVIEWER COMMENTS

Reviewer #2 (Remarks to the Author):

The authors improved the previous version, but there are still quite a few technicalities that in my eyes are not clear or can be improved.

Detailed comment are reported below

Not clear whether in the GEE the authors used the p-values for significance or the corrected q-values. Please, clarify.

What post-hoc analysis they did for the alpha diversity?

I have doubts that the random effect analysis done using MaAsLin2 is appropriate. Why the authors used aged and sex as random effect? Intuitively, I would rather estimate the association of the taxa abundance with the status after accounting for sex and age. This means that the latter need to be included in the model as fixed effects. Can the authors please clarify why they included all other variables as random effects?

Table at pg 11 is wrong. There should be a significant star also between Control and RBD-FDR and between RBD and Early PD. I imagine that this was not reported significant as after multiple testing correction the values increased. Hence, please report the q-values and not the p-values. Also, they say that there is no difference between RBD and Early-PD, but the p-value in the table is sign. In general, I would rather trust the permanova analyses and not necessarily the ANOVA done on the PCoA coordinates. In my opinion this is hard to interpret. In other words what does it mean that two groups have statistically different coordinates along PCoA 1 but not PCoA2? Moreover, considering that the two coordinates explain only ~12% of the data, a lot of the data structure is represented in the other coordinates, so maybe the authors should do similar analysis for the other coordinates as well? I am not sure this is very informative in general. Hence, I will clearly explain and report the data for the Permanova only.

Panel B in the fig 5 at pg 21 is not clear. Is here displayed the R2 of a Permanova test (hence, multivariate analysis) or the R2 of a linear models between factors and taxa? If it is the latter, this is an univariate test, but what taxa was chosen to test this? In the legend they write "univariate analysis of Permanova". I am afraid I do not know what does this mean. Permanova stands for "Permutational multivariate analysis of variance" so it is multivariate. Can the author please clarify.

The ML part is still a bit unclear to me. They split the data in 70% and 30%. They then use the 70% to train the model. As far as I understand they use the 30% left out as an hand out dataset. Then they train a model using the 70%. But report the performance of this model tested on the training dataset as well. Meaning that they had to split the 70% of the data again to obtain a training datasets and a testing dataset. Otherwise how would they test the performances of the model? In my opinion this is not necessary. Because the authors actually test the model twice, 1) using the testing set from the 70%; 2) using the hold out part. The disadvantage of this is that the samples size decreases quite a bit and we know from previous works that this is a major factor affecting model accuracy. If the authors want to test the model on a hold out datasets, they could try to verify how the model perform when the testing dataset is composed only of Controls and RBD-FDR. Theoretically, the model should not predict any RBD here as there are none. The difficulty here is to estimate the accuracy as there are no RBD samples in the testing set. Also, I would find very interesting if the authors verify also whether a robust ML model can be build to differentiate RBD from RBD-FRD. This will show that all other related factors are not affecting the predictive power of the microbiome in identifying RBD.

For these reasons I believe that the paper needs to be further revised before publication

Reviewer #4 (Remarks to the Author):

This is a very interesting study that seeks to begin to understand how a pre-PD population (in this case, those with RBD) may be displaying alterations to the gut microbial community. Overall, this is a very critical study that will help address directionality of the microbiome's relationship and contributions to PD. (ie. does disease manifest first or do particular microbial taxa change first?). The finding that particular taxa that are associated with PD, to be observed altered in the same direction, in those with RBD is quite exciting and important for the field.

I was asked to focus on those comments initially discussed by reviewer #3 in regards to microbiome composition assessment and interpretation. Largely, to me, the reviewers' initial concerns were valid. Particularly, the need to ensure that sex-dependent and drug-dependent effects (that were dissimilar between disease-state groups) was critical. To me, the authors do a very good job addressing these concerns in both the reply and the manuscript. Indeed, others have shown that PD-specific microbiome "signatures" appear even after controlling for diet, geography, and treatments. So, seeing a similar effect in RBD after controlling for these necessary variables is important.

I unfortunately disagree with the reviewer that a "metabolic pathway" analysis is/was necessary. PICRUSt has many caveats that make its interpretation difficult. The authors only know the 16S sequences for certain, and assigning a slew of microbial genes based on this genome region is imprecise. In the discussion surrounding these data ~lines 345, the authors should be careful to mention that this is an inference to genetic content. They did not perform metagenomic sequencing nor metabolomics. These pathways are simply predicted based on 16S taxonomy and so this should be explicitly mentioned and care taken in their inferences to not be over-interpreted.

However, overall, I think the authors did a comprehensive response to the prior reviews and this body of work will be extremely worthwhile to the field.

Reviewer #5 (Remarks to the Author):

**In terms of response to the reviewers, I believe the authors actually made an error in response to reviewer 1, with regards to staging of PD. Specifically:
The study is setup of a spectrum between control to pre-RBD to RBD to PD, with 'pre-RBD' considered to be Stage 0-1. Stage 0-1 is inaccurate; if they have changes in the brain causing mild REM atonia loss, they are stage 2 at least, and if not, they are stage 0 (or maybe 1, although we have no way to know). Moreover:
1. the 'pre-RBD' group should not be considered as a pure 'pre-RBD' sample. It is more likely that it is a mix of true 'pre-RBD' and normal relatives (i.e. people who are relatives and happen to have REM atonia scoring in the high-normal range). So, even if there were no difference between true 'pre-RBD' and RBD patients, combining these two populations essentially guarantees an intermediate spectrum between controls and RBD.
2. The prodromal-RBD subjects are all first-degree relatives of the RBD subjects. As such, could it be that their environment is more similar to the RBD subjects than controls, which could influence their gut microbial environment? Could this be an alternative explanation of the seeming overlap between prodromal-RBD, and RBD/PD metabolic pathways? Certainly this possibility needs to be pointed out**

Other specific comments for consideration

1. Discussion, line 393: "which re-emphasizes the critical role of gut microbiota in the pathogenesis of α -synucleinopathy" and line 478: "Therefore, drugs targeting at constipation and specific microbes at early stages of disease (e.g., RBD and RBD-FDR), will be of great significance for future prevention and disease-modifying therapy of α -synucleinopathy" are both particularly strong statements to make. Although the results of this study are interesting, a clearly causal relationship between changes in the gut microbiome and alpha-synucleinopathies still remains speculative (no mediation analysis in the world can eliminate reverse causality as a possible alternate explanation).

2. Since the analyses are conducted at the genus level, one assumption that must be made is that the microorganisms at each genus have a similar trend of association with disease (e.g. all or most species of a certain genus are reduced in PD, and thus that genus as a whole is shown to be reduced overall). It could, however, be the case that some members are increased while others are decreased evenly, and thus, signals could be lost without having a finer degree of resolution (i.e. species-level).

3. The Results section does not refer to all figures (e.g. Fig 2A) and some are improperly referenced (e.g. Fig "4C" on lines 297, 301 refers to "5C")

Comments and Responses

We greatly appreciated the insightful, helpful, and encouraging comments and suggestions of the reviewers. We have addressed all substantial concerns with regard to statistical analysis and data interpretation in the following point-by-point responses.

REVIEWER COMMENTS

Reviewer #2 (Remarks to the Author):

Comment 1: The authors improved the previous version, but there are still quite a few technicalities that in my eyes are not clear or can be improved. Detailed comment are reported below

Response 1: We thank the reviewer for his/her careful review and insightful comments. We responded to the issues and concerns raised by the reviewer as follows.

Comment 2: Not clear whether in the GEE the authors used the p-values for significance or the corrected q-values. Please, clarify.

Response 2: We thank for the reviewer's comment. In this manuscript, all potential multiple comparisons, including post hoc analyses in the GEE model, were adjusted using the Benjamini-Hochberg method. Thus, false discovery rate (i.e., q-value), rather than p-value, would be reported in the main text accordingly. Based on the reviewer's suggestion, we have made clearer descriptions of GEE in the "Clinical characteristics analysis" section of method (**paragraph 1, page 38**): "A two-sided p value less than 0.05 was considered as statistically significant. For post hoc multiple comparisons in the GEE model (i.e., between each pair of groups), Benjamini-Hochberg False Discovery Rate was used to adjust p values, and a false discovery rate less than 5% (i.e., q-value <0.05) was accepted and indicated statistical significance."

Comment 3: What post-hoc analysis they did for the alpha diversity?

Response 3: The method we used for post-hoc analysis of alpha diversity was the Benjamini-Hochberg False Discovery Rate. In addition, we found that we have made a typo mistake in reporting alpha-diversity data in the Supplementary Table 2. In particular, we erroneously reported p-values <0.001 for all alpha-diversity indices. In fact, none of them were significantly different between groups (i.e., all p-values ≥ 0.05), as we originally reported in the main text (**paragraph 1, page 10**): "We observed that alpha diversity (Chao 1, Gini

Simpson, and Shannon indexes) at genus levels were comparable between groups (Supplementary Table 2).” We have carefully amended the results and added the description of post-hoc analysis in the revised Supplementary Table 2.

Comment 4: I have doubts that the random effect analysis done using MaAsLin2 is appropriate. Why the authors used aged and sex as random effect? Intuitively, I would rather estimate the association of the taxa abundance with the status after accounting for sex and age. This means that the latter need to be included in the model as fixed effects. Can the authors please clarify why they included all other variables as random effects?

Response 4: The reviewer raised the issue of the random effect of MaAsLin2 model and the importance of considering age and sex influences in the differential abundance analysis. In this manuscript, we did not include age and sex as fixed effects because we have separately discussed the effects of all confounders, including age and sex, on microbiota analysis in the "Host factors exert effects on the microbiota at early α -synucleinopathy" section of the Results (**page 19-20, Figure 5**).

However, we agreed with the reviewer that treating age and sex as random effects were probably inappropriate, as both of them are not multi-level categories that include nondependent data (<https://doi.org/10.1017/CBO9780511790942>; <https://doi.org/10.1187/cbe.17-12-0280>). In this revision, we reported two MaAsLin2 models in the part of differential abundance analysis, the first was constructed with four stages (reference = control) as fixed effect (i.e., unadjusted model), and the second including four stages, age, and sex as fixed effects to identify the influences of age and sex on the differential abundance analysis (i.e., adjusted model). The random effect in both models was *family id* to incorporate the variability in family clustering (**page 13-14, Figure 3**). In general, the findings were consistent with previous model, except that genus *Desulfovibrio* only showed marginal increases in RBD and early PD as compared to controls (MaAsLin2, q-value = 0.057 and 0.067, respectively) in the adjusted MaAsLin2 model (revised **Supplementary Table 9**).

Comment 5: Table at pg 11 is wrong. There should be a significant star also between Control and RBD-FDR and between RBD and Early PD. I imagine that this was not reported significant as after multiple testing correction the values increased. Hence, please report the q-values and not the p-values. Also, they say that there is no difference between RBD and Early-PD, but the p-value in the table is sign.

Response 5: The table in Fig.2B at page 11 demonstrates the supplemental results of pairwise comparisons of PERMANOVA tests. This table primarily showed the R^2 values from PERMANOVA, but not the detailed q-values. The full results of PERMANOVA tests could be found in Supplementary Table 3, that there were no significant differences of microbial composition between control and RBD-FDR ($R^2 = 0.0075$, q-value = 0.060) as well as between RBD and early PD ($R^2 = 0.008$, q-value = 0.066). According to the reviewer's comment, we have further added the detailed q-values in this table and modified the legend of Figure 2.

Comment 6: In general, I would rather trust the permanova analyses and not necessarily the ANOVA done on the PCoA coordinates. In my opinion this is hard to interpret. In other words what does it mean that two groups have statistically different coordinates along PCoA 1 but not PCoA2? Moreover, considering that the two coordinates explain only ~12% of the data, a lot of the data structure is represented in the other coordinates, so maybe the authors should do similar analysis for the other coordinates as well? I am not sure this is very informative in general. Hence, I will clearly explain and report the data for the Permanova only.

Response 6: We thank the reviewer's suggestion. In this manuscript, the conclusions on similarities and dissimilarities in microbial composition were drawn based on PERMANOVA test. An additional comparison of PCoA1 and PCoA2 were made due to the heterogeneous dispersion of data between the groups. In other words, PERMANOVA results were sensitive to heterogeneous dispersion in unbalanced study design (*doi: <https://doi.org/10.1890/12-2010.1>*). The significant difference between early PD and control/RBD-FDR may be influenced by the greater dispersion and relatively small size of early PD group. For this reason, we displayed the clustering patterns of each group in ordination plot and examined statistical difference using ANOVA, which could add to the evidence on microbial compositional differences between early PD and control/RBD-FDR groups.

However, as the reviewer pointed out, the ANOVA done on the PCoA coordinates might be limited by the total variations explained by the first two principal components (11.78%). Even when comparing the first nine principal components, the total explained variation was still lower than 30% (27.1%, Supplementary Table 2B). Therefore, in the revised manuscript, we did not highlight the results of principal component comparisons, as suggested by the reviewer. Instead, we have modified Fig. 2B (**page 11-12**) to clearly demonstrate PERMANOVA results as well as the clustering patterns of gut microbiota between different groups, with reference to a previous publication (*doi:10.7717/peerj.2430*).

Comment 7: Panel B in the fig 5 at pg 21 is not clear. Is here displayed the R² of a Permanova test (hence, multivariate analysis) or the R² of a linear models between factors and taxa? If it is the latter, this is an univariate test, but what taxa was chosen to test this? In the legend they write “univariate analysis of Permanova”. I am afraid I do not know what does this mean. Permanova stands for “Permutational multivariate analysis of variance” so it is multivariate. Can the author please clarify.

Response 7: The univariate R² in the figure 5 at page 21 represents the R² from PERMANOVA test. In this manuscript, "univariate R²" refers to the assessment of a single factor (single host factor, e.g., "distance ~ age"), as opposed to the multifactor PERMANOVA test (e.g., "distance ~ four stages + age"). These data demonstrate the effect of each potential confounding factor on the overall microbial composition.

We agreed with the reviewer that “univariate R²” would be confusing in the context of multivariate data analysis. Therefore, we used “PERMANOVA R²” in the revised manuscript, indicating the inter-individual variation explained by each host factor in PERMANOVA test without adjusting four stages or other confounders. And the full description of Figure 5(B) has been modified as follows (**page 22**): “Interactions of overall microbial composition and host factors. Bar plot displayed the inter-individual variation explained by each host factor in whole samples by using PERMANOVA test (i.e., PERMANOVA R², permutations = 99,999). “***”, “**” and “*” represented Benjamini-Hochberg adjusted p values < 0.001, < 0.01, and < 0.05, respectively. Heatmap showed stage-specific (i.e., each stage versus control) impact of host factors. All significant associations were highlighted with red color (q value < 0.05 from the PERMANOVA model). See also Supplementary Table 4.”

Comment 8: The ML part is still a bit unclear to me. They split the data in 70% and 30%. They then use the 70% to train the model. As far as I understand they use the 30% left out as an hand out dataset. Then they train a model using the 70%. But report the performance of this model tested on the training dataset as well. Meaning that they had to split the 70% of the data again to obtain a training datasets and a testing dataset. Otherwise how would they test the performances of the model? In my opinion this is not necessary. Because the authors actually test the model twice, 1) using the testing set from the 70%; 2) using the hold out part. The disadvantage of this is that the samples size decreases quite a bit and we know from previous works that this is a major factor affecting model accuracy. If the authors want to test the model on a hold out datasets, they could try to verify how the model perform when the testing dataset is composed only of Controls and RBD-FDR. Theoretically, the model should not predict any RBD

here as there are none. The difficulty here is to estimate the accuracy as there are no RBD samples in the testing set.

Response 8: We thank the reviewer's comment. Our prior revisions to the ML part have incorporated suggestions from different reviewers, including the issue of feature selection and the generalization of ML model as raised by another reviewer. Considering that this is a single-centre study, one would need to carefully evaluate whether this classification model was also suitable for any unseen data. Therefore, we have applied a standard ML approach, that divide the dataset into three parts, training, validation, and test set (*doi:10.1038/s41467-022-34405-3*; *doi: https://doi.org/10.1016/j.chom.2020.06.004*). Training and validation dataset (i.e., nested cross-validation) was required for feature selection and model fitting, and the test set was used for assessing final model performance in a hold-out dataset.

We agreed with the reviewer that one potential issue with this stringent ML approach is the reduced sample size of training dataset. Therefore, we explored the relationship between training/test set size and model accuracy. The detailed method was depicted in below **Figure 1 (in this reply letter)**, that a subset of data was randomly selected (60%, 70%, 80% or 90%) from the original dataset, followed by random forest-recursive feature elimination (RF-RFE) method (25 repeats of 10-fold cross-validation). The above process was repeated 25 times (i.e., resampling the data subset 25 times) to obtain the distribution of final fit model accuracy in training set as well as the prediction accuracy in hold-out test set (**Table 1 in this reply letter**). Statistical analysis (**Figure 2 in this reply letter**) showed that training with 60, 70, 80, and 90% of original data had comparable model accuracy (0.72 ± 0.05 vs 0.73 ± 0.04 vs 0.74 ± 0.02 vs 0.74 ± 0.02 , Kruskal-Wallis test p-value = 0.57). For the performance in test data, although using 90% of data as the training set showed a better accuracy than 60% (0.67 ± 0.04 vs 0.72 ± 0.07 , q-value = 0.007), a higher variability of accuracy could also be observed as the size of the test data decreases.

Taken together, it appears that using larger training data would not greatly improve model accuracy in this study. In addition, we found that ML process without resampling of training and test data would also yield bias in prediction evaluations. With references to previous microbiome-based classification models (*doi:10.1038/s41467-022-34405-3*; *doi:https://doi.org/10.1016/j.chom.2020.06.004*), we further modified ML method in the

Figure 1 (in this reply letter) Model accuracy for different sizes of training/test data

For each random forest classification model, the original dataset was randomly partitioned into an outer training set (80%) and an independent test set (20%). Within the outer training set, the data were further split into inner training set and validation set for feature selection (“Inner loop”, 25 repeats of 10-fold cross-validation) based on the recursive feature elimination algorithm. The final model was trained with the selected features and its performance were examined in the test set. The whole process of feature selection, final model training, and performance testing was repeated 25 times (i.e., outer loop), and the distribution of final fit model accuracy in training set as well as the prediction accuracy in hold-out test set would be reported.

following parts: 1) splitting the data into 80% training and 20% test, as 80% training data showed almost similar accuracy to 90%, but less variation of accuracy in the hold-out dataset; 2) resampling training/test set 25 times, and reported the average model accuracy and area under curve (AUC); 3) features appearing in $\geq 60\%$ (15/25) of the final fit models were reported as classification features. Moreover, we have added a supplementary figure of ML analysis framework (**Supplementary Figure 5**) and made a clearer description of the ML approach in **page 39-40** as follows: “Cross-validation random forest machine learning algorithm was performed to estimate the accuracy of gut microbiota in discriminating RBD from control and RBD-FDR group. The predictors were CLR transformed abundance of filtered genera (n = 88) as described in the above differential abundance analysis. For each prediction model, the dataset (e.g., control and RBD) was divided into a training set (80% of whole dataset) and a test set (20%). In order to reduce sampling error, the original case/control ratios were preserved in the new datasets via stratified sampling. We applied recursive feature elimination (RFE) algorithm to select predictors from the training set, using the “rfe” function (25 repeats of 10-fold cross-validation) from caret R package (version 6.0-92). The final list of predictors kept in trained model were determined by the prediction accuracy, and the performance of the model was further evaluated in the hold-out test set. The whole process from data splitting, feature selection/model fitting, to prediction evaluation was repeated 25 times (Supplementary Fig. 5). Receiver operating characteristic (ROC) curves and the area under the curve (AUC) were calculated with R package “pROC”.”

Figure 2 (in this reply letter) Comparisons of model accuracy for different sizes of data

The differences of model accuracy between groups were compared using Kruskal-Wallis test with post-hoc analysis.

Table 1 (in this reply letter) Detailed average model accuracy at each resampling

	Mean model accuracy with selected feature set (25 repeats of 10-fold CV)							
	60%		70%		80%		90%	
	Final	Test	Final	Test	Final	Test	Final	Test
Resample 01	0.789	0.607	0.725	0.682	0.770	0.714	0.752	0.607
Resample 02	0.663	0.670	0.710	0.682	0.725	0.679	0.724	0.786
Resample 03	0.741	0.634	0.762	0.682	0.716	0.679	0.748	0.786
Resample 04	0.759	0.688	0.715	0.612	0.766	0.661	0.712	0.750
Resample 05	0.639	0.679	0.751	0.694	0.743	0.607	0.744	0.821
Resample 06	0.735	0.696	0.736	0.682	0.739	0.679	0.748	0.714
Resample 07	0.693	0.705	0.720	0.765	0.743	0.696	0.704	0.643
Resample 08	0.759	0.688	0.736	0.635	0.730	0.750	0.724	0.714
Resample 09	0.741	0.670	0.705	0.718	0.739	0.661	0.756	0.679
Resample 10	0.542	0.625	0.570	0.718	0.752	0.732	0.764	0.786
Resample 11	0.771	0.661	0.736	0.647	0.766	0.679	0.740	0.571
Resample 12	0.705	0.661	0.767	0.671	0.748	0.661	0.740	0.750
Resample 13	0.759	0.705	0.777	0.671	0.761	0.679	0.756	0.786
Resample 14	0.717	0.679	0.751	0.694	0.757	0.714	0.752	0.714
Resample 15	0.771	0.670	0.720	0.718	0.698	0.714	0.748	0.607
Resample 16	0.693	0.670	0.736	0.624	0.743	0.732	0.724	0.714
Resample 17	0.735	0.714	0.736	0.706	0.739	0.696	0.768	0.821
Resample 18	0.729	0.580	0.679	0.682	0.752	0.643	0.748	0.679
Resample 19	0.747	0.732	0.777	0.659	0.698	0.679	0.744	0.714
Resample 20	0.699	0.696	0.731	0.729	0.766	0.625	0.728	0.714
Resample 21	0.711	0.634	0.756	0.659	0.716	0.714	0.740	0.643
Resample 22	0.687	0.714	0.767	0.659	0.748	0.661	0.744	0.714
Resample 23	0.735	0.643	0.741	0.694	0.748	0.607	0.756	0.714
Resample 24	0.777	0.661	0.674	0.694	0.721	0.750	0.720	0.786
Resample 25	0.777	0.688	0.731	0.671	0.739	0.625	0.708	0.786
Mean	0.72	0.67	0.73	0.68	0.74	0.68	0.74	0.72
SD	0.05	0.04	0.04	0.03	0.02	0.04	0.02	0.07

Details of training and test model accuracies from each resample. SD, standard deviation.

Comment 9: Also, I would find very interesting if the authors verify also whether a robust ML model can be build to differentiate RBD from RBD-FRD. This will show that all other related factors are not affecting the predictive power of the microbiome in identifying RBD. For these reasons I believe that the paper needs to be further revised before publication.

Response 9: We appreciate the reviewer's suggestion. In the revised manuscript, we have reported the ML model for the classification of RBD and RBD-FDR (**Fig. 4D, paragraph 2, page 17**). The data showed that microbial markers had a comparable ability in discriminating RBD from control and RBD-FDR, albeit there were some emerging RBD/PD-like microbial changes at RBD-FDR as compared with controls (i.e., *[Eubacterium]_ventriosum_group* and *Collinsella*).

Reviewer #4 (Remarks to the Author):

Comment 1: This is a very interesting study that seeks to begin to understand how a pre-PD population (in this case, those with RBD) may be displaying alterations to the gut microbial community. Overall, this is a very critical study that will help address directionality of the microbiome's relationship and contributions to PD. (ie. does disease manifest first or do particular microbial taxa change first?). The finding that particular taxa that are associated with PD, to be observed altered in the same direction, in those with RBD is quite exciting and important for the field.

Response 1: We sincerely express our gratitude for the positive responses to our manuscript and the concise summary of our work. We responded to the reviewer's comments point-by-point as follows.

Comment 2: I was asked to focus on those comments initially discussed by reviewer #3 in regards to microbiome composition assessment and interpretation. Largely, to me, the reviewers initial concerns were valid. Particularly, the need to ensure that sex-dependent and drug-dependent effects (that were dissimilar between disease-state groups) was critical. To me, the authors do a very good job addressing these concerns in both the reply and the manuscript. Indeed, others have shown that PD-specific microbiome "signatures" appear even after controlling for diet, geography, and treatments. So, seeing a similar effect in RBD after controlling for these necessary variables is important.

Response 2: We thank the reviewer for acknowledging that our replies and revisions have addressed the issues raised by reviewer #3. These valuable comments and suggestions, especially on host-microbiome interactions, have helped us to greatly improve the quality of this manuscript.

Comment 3: I unfortunately disagree with the reviewer that a "metabolic pathway" analysis is/was necessary. PICRUSt has many caveats that make its interpretation difficult. The authors only know the 16S sequences for certain, and assigning a slew of microbial genes based on this genome region is imprecise. In the discussion surrounding these data ~lines 345, the authors should be careful to mention that this is an inference to genetic content. They did not perform metagenomic sequencing nor metabolomics. These pathways are simply predicted based on 16S taxonomy and so this should be explicitly mentioned and care taken in their inferences to not be over-interpreted.

Response 3: We agreed that the metabolic changes predicted from the 16S sequence data should be carefully interpreted. In this regard, we have rephrased the beginning of the discussion section on metabolic pathways (**paragraph 2, page 30**): “In this study, the functional profile of gut microbiota was predicted from 16S rRNA gene sequencing data, suggesting increased fatty acids metabolism, and decreased biosynthesis of preQ0 and vitamin B₁₂, at early α -synucleinopathy”. Accordingly, we have also revised the limitation part (**paragraph 2, page 32**) as follows: “Finally, this study is limited to the compositional profile of gut microbiota and functional prediction based on 16S rRNA gene sequencing data, a comprehensive survey of gut microbiota (at species- and strain-levels) and gut metabolism by metagenomics and metabolomics are needed in the subsequent studies.”

Comment 4: However, overall, I think the authors did a comprehensive response to the prior reviews and this body of work will be extremely worthwhile to the field.

Response 4: We thank the reviewer again for supporting the potential interest of our findings.

Reviewer #5 (Remarks to the Author):

Comment 1: In terms of response to the reviewers, I believe the authors actually made an error in response to reviewer 1, with regards to staging of PD. Specifically: The study is setup of a spectrum between control to pre-RBD to RBD to PD, with ‘pre-RBD’ considered to be Stage 0-1. Stage 0-1 is inaccurate; if they have changes in the brain causing mild REM atonia loss, they are stage 2 at least, and if not, they are stage 0 (or maybe 1, although we have no way to know).

Response 1: We agree with the reviewer that it is difficult to accurately assign clinical cases to specific pathological stages until definitive pathological evidence is available. This will be even more difficult when many of the clinical symptoms are nonspecific (e.g., constipation and olfactory dysfunction). In this manuscript, we classified RBD-FDR as stage 0-1 instead of stage 0-2 with the following considerations:

- 1) When comparing PD prodromal markers, RBD-FDR showed a significantly lower total likelihood ratio of prodromal PD than RBD patients (0.46 ± 0.55 vs 1.4 ± 0.98 , q -value < 0.001 , Supplementary Table 1), suggesting that RBD-FDR exhibit much milder and/or less diffuse alpha-synuclein pathology than v-PSG confirmed RBD. In fact, the overall likelihood ratio of prodromal PD in RBD-FDR was more similar to that of controls (0.46 ± 0.55 vs 0.58 ± 0.72 , q -value = 0.27). Therefore, we considered that RBD-FDR may not reach Braak stage 2, or at least not at the same level of RBD;
- 2) As for the “mild REM sleep atonia loss” mentioned by the reviewer, we only included RBD-FDR without v-PSG confirmed RBD (i.e., unaffected RBD-FDR) in this manuscript. Although these RBD-FDR might have higher level of REM sleep without atonia (RSWA) than FDRs of controls, their chin muscle activity were still within the normal range (i.e., lower than the diagnostic cut-off - chin muscle activity = 18%) (*doi: doi:10.1002/ana.25435*). To our knowledge, there is no evidence that high-normal range RSWA is associated with the lesions in brainstem nucleus (e.g., locus coeruleus/subcoeruleus) that controlling muscle atonia during REM sleep.

However, as astutely pointed out by the reviewer in the comment 2, RBD-FDR is probably an intermediate spectrum between controls and RBD. In other words, some RBD-FDR are highly likely to convert to full-blown RBD and neurodegeneration, while others are not at higher risk than the general population. This concept of spectrum may not be limited to RBD-FDR, but also applies to RBD and control groups. Therefore, in the revised Figure 1 and **Figure 3 in this reply letter**, we refer to controls as Braak stage 0-1 (probably consists of incidental alpha-synuclein pathology) and RBD-FDR as stage 0-2, while RBD represents

stage 2-3 (some presenting with subtle parkinsonism, suggesting lesions in midbrain especially substantia nigra), and early PD corresponds to Braak stage 3-4 (relative intact cognitive function, indicating no severe involvement of neocortical areas)

(doi:10.1016/s0197-4580(02)00065-9).

Figure 3 A staging model of early α -synucleinopathy

We proposed a clinical staging model of early α -synucleinopathy, which represent the pathological staging of Parkinson's disease (i.e., Braak staging). Four different clinical stages were controls (Braak stage 0-1), RBD-FDR (stage 0-2), patients with RBD (stage 2-3) and early PD (stage 3-4). Abbreviations: RBD, REM sleep behavior disorder; RBD-FDR, first-degree relatives of patients with RBD; PD, Parkinson's disease; ND, neurodegenerative disease; LR, likelihood ratio.

Comment 2: Moreover: 1. the ‘pre-RBD’ group should not be considered as a pure ‘pre-RBD’ sample. It is more likely that it is a mix of true ‘pre-RBD’ and normal relatives (i.e. people who are relatives and happen to have REM atonia scoring in the high-normal range). So, even if there were no difference between true ‘pre-RBD’ and RBD patients, combining these two populations essentially guarantees an intermediate spectrum between controls and RBD.

Response 2: As elaborated above in the response to the comment 1, we fully agreed that RBD-FDR is a mixed group of "pre-RBD" and normal relatives. In the revised manuscript, we refer to RBD-FDR as a “potentially high-risk group for RBD and PD” or “susceptible individuals at a much earlier stage of α -synucleinopathy than RBD patients”, rather than explicitly describing it as "pre-RBD".

Comment 3: 2. The prodromal-RBD subjects are all first-degree relatives of the RBD subjects. As such, could it be that their environment is more similar to the RBD subjects than controls, which could influence their gut microbial environment? Could this be an alternative explanation of the seeming overlap between prodromal-RBD, and RBD/PD metabolic pathways? Certainly this possibility needs to be pointed out

Response 3: We fully agreed that shared environment may potentially explain the similar profile of gut microbiota. In fact, when designing this study, in order to minimize the influence of shared environmental factors, we excluded RBD relatives who were cohabiting with RBD probands. In addition, we further adjusted for family clustering in majority of the analyses. For example, *family id* was treated as a cluster factor in GEE model for clinical variables analysis, and as a random effect in MaAsLin for differential microbe analysis.

In this manuscript, there were 48 RBD/early PD patients and 81 RBD-FDR from the same family. To further clarify whether RBD and their family relatives showed similar changes of gut microbes, we performed correlation analysis between 48 pairs of RBD and RBD-FDR (if there were more than one RBD-FDR in the same family, only the one with age and sex closest to the RBD patient was retained). The results showed no significant co-occurrences of two differential genera (i.e., *Collinsella* and [*Eubacterium*]*_ventriosum_group*) between 48 pairs of RBD and RBD-FDR (**Figure 4 in this reply letter**), suggesting that the shared early life environment probably had minor impact, if any, on gut microbiota disturbances identified at early α -synucleinopathy. However, other factors, such as genetics and similar dietary habits, may also affect the gut microbiota in unpredictable ways (*doi:10.1186/s13073-021-01005-7*; *doi:10.1016/j.cgh.2018.08.067*). Taken together, we incorporated the reviewer’s suggestions and discussed this issue in the Limitation part of revised manuscript (**paragraph**

2 page 32): “Also, the similar changes of gut microbes in RBD/early PD and their relatives (RBD-FDR) might be affected by other unmeasured factors, such as genetics, shared dietary habit and early-life exposure. Future analyses incorporating these factors may help understanding the development of gut dysbiosis at early α -synucleinopathy.”

Figure 4 (in this reply letter). Spearman correlation analysis showed no significant relationship between RBD patients and their family relatives (total 48 pairs).

Comment 4: Other specific comments for consideration 1. Discussion, line 393: “which re-emphasizes the critical role of gut microbiota in the pathogenesis of α -synucleinopathy” and line 478: “Therefore, drugs targeting at constipation and specific microbes at early stages of disease (e.g., RBD and RBD-FDR), will be of great significance for future prevention and disease-modifying therapy of α -synucleinopathy” are both particularly strong statements to make. Although the results of this study are interesting, a clearly causal relationship between changes in the gut microbiome and alpha-synucleinopathies still remains speculative (no mediation analysis in the world can eliminate reverse causality as a possible alternate explanation).

Response 4: We appreciate the reviewer’s comments and fully agreed that a causal relationship could not be drawn from this cross-sectional study. Our statement would like to highlight that our findings added to the evidence of previous preclinical and interventional studies that have supported the critical role of gut microbiota in the pathogenesis of Parkinson’s disease (*doi:10.1016/j.cell.2016.11.018*; *doi:https://doi.org/10.1016/j.bbi.2018.02.005*; *doi:10.1212/wnl.0000000000010998*). In the revised manuscript, we have toned down the statement as follows: 1) “which emphasizes the potential role of gut microbiota in the pathogenesis of α -synucleinopathy” (**paragraph 1, page 28**); 2) “Hence, interventions on constipation and specific microbes at early stages of disease (e.g., RBD and RBD-FDR), may be a promising strategy for future prevention and disease-modifying therapy of α -synucleinopathy” (**paragraph 1, page 32**).

Comment 5: 2. Since the analyses are conducted at the genus level, one assumption that must be made is that the microorganisms at each genus have a similar trend of

association with disease (e.g. all or most species of a certain genus are reduced in PD, and thus that genus as a whole is shown to be reduced overall). It could, however, be the case that some members are increased while others are decreased evenly, and thus, signals could be lost without having a finer degree of resolution (i.e. species-level).

Response 5: We agreed with the reviewer that a higher resolution would not only improve the certainty of the findings, but also provides more specific targets for further downstream investigations (e.g., probiotics development). However, taxonomic classification in this study was based on V3-V4 16S rRNA regions, not full-length 16S rRNA or whole gene sequencing, which showed limited accuracy in predicting taxonomy at species-levels ([doi:10.7717/peerj.4652](https://doi.org/10.7717/peerj.4652); [doi:10.1186/s40168-022-01295-y](https://doi.org/10.1186/s40168-022-01295-y)).

In our own data, less than half of the ASV sequences could be assigned to species if using traditional pipeline (naïve Bayes taxonomy classifier against the SILVA v138 99% 16S rRNA databases) as described in the method section of this manuscript (**paragraph 1, page 37**). Therefore, we further applied another pipeline “Pplacer” to improve the prediction of species by maximizing phylogenetic likelihood. The data showed that among differential genera (n = 16, after adjusting multiple confounders), ASV sequences under seven genera could only be reliably (confidence > 90%) grouped into one species, thus, we did not perform further analysis on these genus/species. For the remaining nine genera with more than one species recognized, the data was depicted in the below **Table 2 (in this reply letter)**. In general, similar trends of changes at genus- and species-levels were observed at early stages of synucleinopathy by using Mann-Kendall trend test, except for one species *Butyricicoccus.pullicaeorum* which seemingly changed in the opposite direction to the genus *Butyricicoccus*.

In brief, we shared the concern about the overall accuracy of species identification based solely on single nucleotide polymorphism in hypervariable regions of 16S rRNA ([doi:10.1186/s40168-022-01295-y](https://doi.org/10.1186/s40168-022-01295-y)) and further comprehensive study of gut microbiota (at species- and strain-levels) and gut metabolism by metagenomics and metabolomics are needed. Nevertheless, some preliminary analyses in this reply letter showed similar trends of changes at this higher resolution data. In the revised manuscript, we have added the issue raised by the reviewer in the Limitation section as follows (**paragraph 2 page 32**): “Finally, this study is limited to the compositional profile of gut microbiota and functional prediction based on 16S rRNA gene sequencing data, a comprehensive survey of gut microbiota (at species- and strain-levels) and gut metabolism by metagenomics and metabolomics are needed in the subsequent studies.”

Table 2 (in this reply letter) Trends of changes at species-levels

	Relative abundance of differential genus and species								Mann-Kendall trend test	
	Control		RBD-FDR		RBD		Early PD		Coefficient	P value
	Mean	SD	Mean	SD	Mean	SD	Mean	SD		
Collinsella	0.1282	0.2964	0.2744	0.7477	0.4751	1.1882	0.4208	0.8019	0.130	0.001
Collinsella.aerofaciens	0.1223	0.2957	0.2709	0.7466	0.4538	1.1854	0.4183	0.8031	0.123	0.002
Collinsella.intestinalis	0.0058	0.0433	0.0014	0.0163	0.0155	0.1349	0.0025	0.0147	0.028	0.52
Collinsella.tanakaei	0.0000	0.0000	0.0021	0.0165	0.0026	0.0294	0.0000	0.0000	0.018	0.68
Collinsella.stercoris	0.0000	0.0000	0.0000	0.0000	0.0010	0.0134	0.0000	0.0000	0.036	0.41
Christensenellaceae_R-7_group	0.2645	0.5819	0.2145	0.6042	0.6647	1.5711	1.0874	1.4778	0.137	0.001
Christensenella.timonensis	0.1399	0.3746	0.1510	0.5364	0.4765	1.3514	0.8107	1.2875	0.135	0.001
Gracilibacter.thermotolerans	0.1051	0.2822	0.0498	0.1790	0.1054	0.3111	0.0831	0.1446	0.023	0.58
Lutispora.thermophila	0.0000	0.0000	0.0002	0.0023	0.0010	0.0092	0.0051	0.0202	0.113	0.01
Anaerofustis.stercorihominis	0.0005	0.0054	0.0000	0.0000	0.0000	0.0000	0.0012	0.0073	0.009	0.84
Butyricoccus	0.6446	0.7458	0.6054	0.6473	0.3726	0.6665	0.2539	0.3565	-0.219	<0.001
Agathobaculum.butyriciproducens	0.5243	0.6519	0.4969	0.5163	0.3251	0.5782	0.2020	0.3201	-0.189	<0.001
Acetitomaculum.ruminis	0.0705	0.2297	0.0627	0.2634	0.0306	0.1507	0.0071	0.0427	-0.098	0.03
Agathobaculum.desmolans	0.0497	0.1131	0.0458	0.1486	0.0159	0.0601	0.0217	0.0667	-0.109	0.01
Catonella.morbi	0.0000	0.0000	0.0000	0.0000	0.0010	0.0134	0.0000	0.0000	0.036	0.41
Butyricoccus.pullicaeorum	0.0000	0.0000	0.0000	0.0000	0.0000	0.0000	0.0230	0.1279	0.11	0.02
Family_XIII_AD3011_group	0.0387	0.0981	0.0228	0.0766	0.1040	0.2154	0.1179	0.1993	0.174	<0.001
Aminicella.lysinilytica	0.0010	0.0099	0.0000	0.0000	0.0102	0.1017	0.0000	0.0000	0.038	0.39
Eubacterium.brachy	0.0040	0.0191	0.0049	0.0282	0.0264	0.1033	0.0131	0.0433	0.11	0.01
Eubacterium.saphenum	0.0040	0.0314	0.0031	0.0266	0.0079	0.0430	0.0000	0.0000	0.019	0.66
Lachnospira	0.4337	0.5887	0.4415	0.8904	0.1990	0.4458	0.2124	0.4496	-0.171	<0.001
Lachnospira.pectinoschiz	0.0982	0.3435	0.1039	0.3910	0.0604	0.2596	0.0419	0.1957	-0.048	0.26

Lachnospira.multipara	0.0257	0.0963	0.0558	0.2468	0.0203	0.1181	0.0571	0.2533	-0.039	0.37
Lactobacillus.rogosae	0.0017	0.0177	0.0000	0.0000	0.0000	0.0000	0.0000	0.0000	-0.061	0.17
Oscillospiraceae_UCG-005	0.1432	0.3709	0.1078	0.3257	0.2933	0.5686	0.4898	0.7295	0.171	<0.001
Sporobacter.termitidis	0.1127	0.3037	0.0837	0.2967	0.1940	0.4533	0.3249	0.5167	0.11	0.01
Intestinimonas.massiliensis	0.0000	0.0000	0.0049	0.0392	0.0014	0.0115	0.0000	0.0000	0.032	0.46
Solibaculum.mannosilyticum	0.0000	0.0000	0.0010	0.0108	0.0000	0.0000	0.0000	0.0000	-0.018	0.69
Clostridia_UCG-014	0.2297	0.7264	0.1985	0.4742	0.7613	2.0671	1.1501	2.0259	0.148	<0.001
Vallitalea.pronyensis	0.1072	0.3423	0.1060	0.3334	0.3542	0.9478	0.4086	0.8024	0.122	0.003
Ruminococcoides.bili	0.0008	0.0079	0.0000	0.0000	0.0000	0.0000	0.0000	0.0000	-0.061	0.17
Cellulosilyticum.ruminicola	0.0006	0.0065	0.0002	0.0023	0.0097	0.0670	0.0000	0.0000	0.040	0.37
Anaerostipes.hadrus	0.0000	0.0000	0.0000	0.0000	0.0033	0.0370	0.0000	0.0000	0.051	0.24
Uncultured_Ruminococcaceae_g071	0.0315	0.1268	0.0529	0.2528	0.0760	0.1892	0.1045	0.1898	0.141	0.001
Acetanaerobacterium.elongatum	0.0150	0.1072	0.0124	0.0776	0.0379	0.1319	0.0389	0.1086	0.127	0.003
Harryflintia.acetispora	0.0040	0.0175	0.0036	0.0136	0.0059	0.0229	0.0039	0.0165	-0.010	0.82
Anaeromassilibacillus.senegalensis	0.0008	0.0081	0.0050	0.0563	0.0070	0.0396	0.0000	0.0000	0.067	0.13
Phoceamassiliensis	0.0002	0.0021	0.0000	0.0000	0.0000	0.0000	0.0000	0.0000	-0.061	0.17
Clostridium.merdae	0.0000	0.0000	0.0000	0.0000	0.0010	0.0127	0.0000	0.0000	0.036	0.41
Gemmiger.formicilis	0.0000	0.0000	0.0007	0.0080	0.0000	0.0000	0.0000	0.0000	-0.018	0.69
Desulfovibrio	0.0706	0.2170	0.0472	0.1652	0.1038	0.2414	0.1698	0.3612	0.124	0.003
Desulfovibrio.piger	0.0686	0.2170	0.0400	0.1602	0.0763	0.2213	0.1117	0.2413	0.062	0.14
Desulfovibrio.legallii	0.0020	0.0170	0.0019	0.0166	0.0084	0.0925	0.0000	0.0000	-0.018	0.68
Desulfovibrio.desulfuricans	0.0000	0.0000	0.0009	0.0074	0.0009	0.0117	0.0017	0.0103	0.037	0.40
Desulfovibrio.simplex	0.0000	0.0000	0.0000	0.0000	0.0007	0.0087	0.0000	0.0000	0.036	0.41

Species under each genus were identified with pipeline “Pplacer”. Only species with a classification confidence > 90% were reported. P values were calculated using Mann-Kendall trend test, and bold values denote statistical significance at the $p < 0.05$ level.

Comment 6: 3. The Results section does not refer to all figures (e.g. Fig 2A) and some are improperly referenced (e.g. Fig “4C” on lines 297, 301 refers to “5C”)

Response 6: We appreciate the reviewer's suggestions and have revised the inappropriate references to the figures accordingly.

REVIEWERS' COMMENTS

Reviewer #2 (Remarks to the Author):

The author explained in details their changes and work and clarified my concerns.

Reviewer #5 (Remarks to the Author):

Comments have been addressed satisfactorily

Comments and Responses

****REVIEWER COMMENTS****

Replies to reviewers:

Reviewer #2 (Remarks to the Author):

Comment: The author explained in details their changes and work and clarified my concerns.

Response: We would like to thank the reviewer for the insightful and helpful comments.

Reviewer #5 (Remarks to the Author):

Comment: Comments have been addressed satisfactorily

Response: We sincerely express our gratitude for the positive responses to our manuscript and the concise summary of our work.